# Recent Developments in Nanocatalyzed Green Synthetic Protocols of Biologically Potent Diverse O-Heterocycles—A Review

Suresh Kumar [1,*], Bhavna Saroha [1], Gourav Kumar [1], Ekta Lathwal [1,2], Sanjeev Kumar [1], Badri Parshad [3], Meena Kumari [4], Naveen Kumar [5], Mabel M. Mphahlele-Makgwane [6,*] and Peter R. Makgwane [7,8]

1 Department of Chemistry, Kurukshetra University, Kurukshetra 136119, Haryana, India; bhavnasaroha0712@gmail.com (B.S.); gouravbura1994@gmail.com (G.K.); ektalathwal@gmail.com (E.L.); sanjeevkumarsahrawat@gmail.com (S.K.)
2 Department of Chemistry, Govt. College, Tigaon, Faridabad 121101, Haryana, India
3 Wellman Center for Photomedicine, Massachusetts General Hospital, Harvard Medical School, Charlestown, MA 02129, USA; bparshad@mgh.harvard.edu
4 Department of Chemistry, Govt. College for Women, Badhra, Charkhi Dadri 127306, Haryana, India; jakhar7meena@gmail.com
5 Department of Chemistry, Maharishi Dayanand University, Rohtak 124001, Haryana, India; naveenkumar.chem@mdurohtak.ac.in
6 Department of Water and Sanitation, University of Limpopo, Private Bag X1106, Sovenga 0727, South Africa
7 Centre for Nanostructures and Advanced Materials, Council for Scientific and Industrial Research (CSIR), Pretoria 0001, South Africa; pmakgwane@csir.co.za
8 Department of Chemistry, University of the Western Cape, Bag X17, Robert Sobukwe Drive, Bellville 7535, South Africa
* Correspondence: suresh_dua47@rediffmail.com (S.K.); mabel.mphahlele-makgwane@ul.ac.za (M.M.M.-M.)

**Abstract:** The dynamic growth in green organic synthetic methodologies for diverse heterocyclic scaffolds has substantially contributed to the field of medicinal chemistry over the last few decades. The use of hybrid metal nanocatalysts (NCs) is one such benign strategy for ensuring the advancement of modern synthetic chemistry by adhering to the principles of green chemistry, which call for a sustainable catalytic system that converts reacting species into profitable chemicals at a faster rate and tends to reduce waste generation. The metal nanoparticles (NPs) enhance the exposed surface area of the catalytic active sites, thereby making it easier for reactants and metal NCs to have an effective interaction. Several review articles have been published on the preparation of metal NCs and their uses for various catalytic heterocyclic transformations. This review will summarize different metal NCs for the efficient green synthesis of various O-heterocycles. Furthermore, the review will provide a concise overview of the role of metal NCs in the synthesis of O-heterocycles and will be extremely useful to researchers working on developing novel green and simple synthetic pathways to various O-heterocyclic-derived molecules.

**Keywords:** nanocatalyst; green chemistry; chalcone; coumarin; furan; oxazole; pyran

## 1. Introduction

Chemistry is an undeniably important part of our daily lives, and its advancements have been developed to provide people with access to innovative medicines for healthier living conditions. Several heterocycle scaffolds have been found to be pharmaceutically active for applications in the development and formulation of new drugs [1]. Heterocycles are found in both natural and synthesized biologically active compounds [2]. Despite incidents [3] such as the clioquinol tragedy, the thalidomide tragedy, and others, the importance of heterocycles in drug discovery continues to rise due to their selectively pharmacophore binding potential. The heterocyclic skeleton can be found in a large number of commercially accessible Food and Drug Administration (FDA)-approved synthesized

drugs [4] (Figure 1). O-containing heterocycles have been shown to be the physiologically most decorative and potent heterocycles [5–7]. Never-ending O-heterocyclic compounds have served as active agents such as antioxidants [8], anticancer agents [9], anti-Alzheimer agents [10,11], antimalarial agents [12], etc.

**Figure 1.** FDA-approved heterocyclic drugs.

The ubiquitous importance of heterocyclic scaffolds motivates researchers to develop a sustainable synthetic path to replace the archaic classical methods. Sustainable chemistry is a trending concept that aims to use effective chemical methodologies for heterocycles synthesis [13,14] and to reduce the cost incurred in drug-designing programs. Though various catalytic synthetic protocols are well established and capable of achieving the required framework, they have a number of drawbacks. For instance, to synthesize coumarin, various Lewis acids and inorganic reagents such as $POCl_3$, $P_2O_5$, PPA, $CF_3COOH$, montmorilonite, zeolite H-BEA, and $W/ZrO_2$ were traditionally used [15], but the primary disadvantages of the conventional techniques are the harsh reaction conditions, non-reusable catalyst, difficult separation, and long reaction time. As a result, the shortcomings of traditional catalysts necessitate the development of a new catalytic system that not only exhibits high activity and selectivity, but also undergoes facile separation from products and great recovery. Nanoparticle-based catalytic systems can be used to achieve the aforementioned goals. In the recent era, nanotechnology and nanoscience in drug design have proven to be the most compatible [16,17] and precise way to achieve environmentally benign methodology [18]. Due to the relevance of nanotechnology in organic syntheses, scientists have developed a growing interest in expanding the expertise of metal NCs to make modern organic transformations more accessible and benign [19].

Nanoparticles as an asset in modern organic synthesis:

The history of nanoparticles (NPs), particularly gold NPs, dates to the fifty-first century AD, when people in China and Egypt utilized gold NPs for therapeutic and aesthetic purposes and artists for manufacturing and adorning glasses. At the beginning of the seventeenth century, the first literature report on colloidal metal NPs was published [20]. In the mid-nineteenth century, Faraday produced the first scientific paper on the effect of light on metal NPs [21]. The use of metal NPs as catalysts was first reported in 1941 when Pd and Pt NPs were used for hydrogenation reaction [22]. Later, with time, a massive breakthrough appeared in the world of metal NPs catalysis. Now, metal NPs as catalysts are assumed to be an enabling innovation in developing chemical transformations that lies at the core of valuable synthetic conventions [23]. A catalyst, as the name implies, is the foreign material that enables a chemical transformation with lower activation energy and can be recovered without any chemical change at the end of the reaction process [24]. The NP-based catalytic system supports the findings of Paul Anastas and John C. Warner's principles of green chemistry [25] (Figure 2). The fundamental principle of green chemistry is to develop novel catalytic systems that may address the challenges of conventional protocols by being both cost-effective and efficient. Low production cost, high stability, strong selectivity, facile separation, and good recyclability are the desirable properties that a green catalyst must possess, and NCs possess many of these. Therefore, the use of metal NPs as a heterogeneous solid catalyst in modern organic synthesis is an effective strategy for achieving the objectives of green chemistry, such as selectivity, easy separation from the reaction mixture, cost-effectiveness, reusability, and many more [26,27]. Thus, the metal NC framework allows chemical transformations to occur quickly and specifically with higher yield in short time intervals, as NCs significantly increase the contact between reactants and catalyst on account of highly nanoscale-exposed active surface sites compared to the bulk.

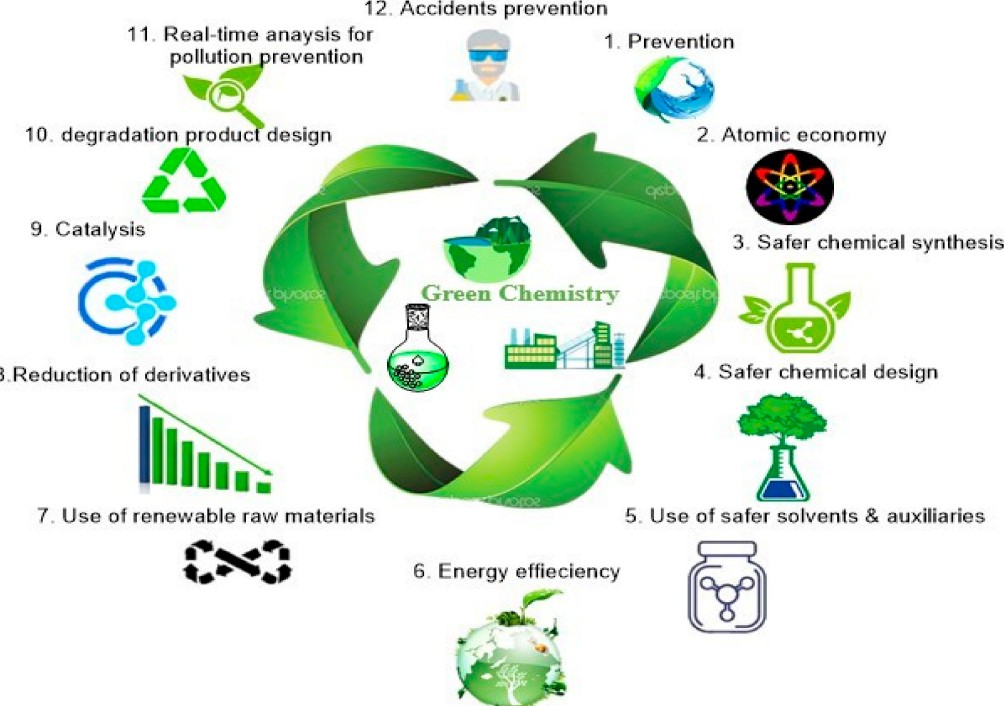

**Figure 2.** Warner's principles of green chemistry.

The chemistry of precious metals has emerged as an interesting prospect for advanced application in catalysis that can be affected by carefully regulating chemical inertness. As a result, numerous metal NPs have been doped with abundant metal oxides to create advanced nanohybrids [28,29] whose synergistic properties lead to increased selectivity and reactivity in diverse chemical processes [30]. The selectivity and reactivity of metal NCs,

which are directly dependent on the morphology, size, crystal structure [31], and surface content of the NC [32], play a crucial role in achieving effective organic transformation. As a result, one can design a metal NC with active sites to achieve high selectivity [33] and robustness by synthesis-controlled fine-tuning of its structural properties in terms of composition, shape, and size. These benefits will allow research to become more resource-efficient, consume less energy, produce less waste and will reduce the environmental effect of traditional synthetic methodologies. Thus, the current nanocatalyzed methods have tremendous potential for future breakthroughs in the synthesis of diverse heterocyclic systems. Besides synthetic chemistry, metal NCs are used in a wide range of applications [34], such as solid rocket propellant [35], in fuel cell applications [36], as thin-film solar cells [37], as biofuel additives [38], and many others [39] (Figure 3). As a result, traditional notions about metals, such as their inertness to chemical processes, have been utterly disproved, and new avenues for interesting usage of metal NPs as a catalyst have opened up when compared to their bulk counterparts' form, such as the inactive bulk form of gold metal catalysts.

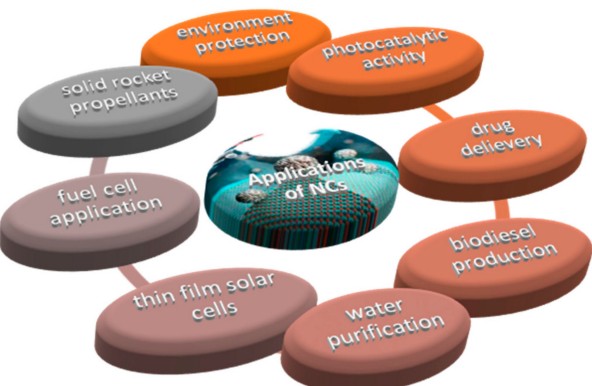

**Figure 3.** Scopes of metal nanocatalysts.

Over the past few decades, metal nanocatalyzed organic transformations have appeared as a prevalent topic [40], and a number of review articles on metal nanocatalyzed heterocyclic synthesis have been published to date [41–46] (Figure 4). Unlike other reviews, the present review concentrates on diverse organic transformations, with an emphasis on O-heterocycles derivatives in terms of green chemistry principles. Here, we offer a brief overview of the green synthesis of biologically active O-heterocycles such as coumarins, oxazoles, chalcones, furans, and pyrans. Researchers can obtain important information regarding the applications of metal NCs in diverse O-heterocyclic syntheses, as in the present review, we have incorporated nanocatalyzed heterocyclic synthesis and its mechanisms (wherever available).

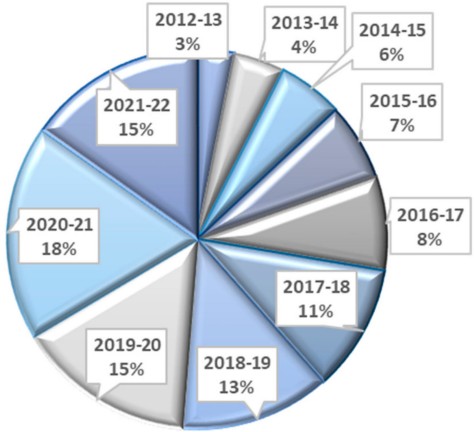

**Figure 4.** Number of review articles on nanocatalyzed heterocyclic synthesis.

## 2. Nanocatalyzed Green Synthesis of Various O-Heterocycles

### 2.1. Synthesis of Furans

Furan is a colorless, flammable, highly volatile liquid that is rapidly and extensively absorbed by the intestine and the lung [47]. Several synthetic and natural furans derivatives have been reported with favorable biological functions, including antihyperglycemic [48], antioxidant [49], anticonvulsant [50], and antiproliferative [51] activities, among various others [52]. Some of the recent nanocatalyzed syntheses of furan derivatives are:

Ji et al. created a new NC based on Pd NPs and N,O-dual-doped hierarchical porous carbon obtained from natural biomass [53]. Under copper-free conditions, the suggested NC was used for a one-pot tandem synthesis of 2-benzofurans (Scheme 1). The substrate scope data revealed that under the specified reaction situation, the substrate with o-substituent was virtually less reactive than the substrates with m- and p-substituents; this could be attributed to the steric effect. The nanocatalyst's large surface area allowed for consecutive intermolecular Sonogashira cross-coupling followed by cyclization in a copper- and ligand-free environment, making the process highly efficient.

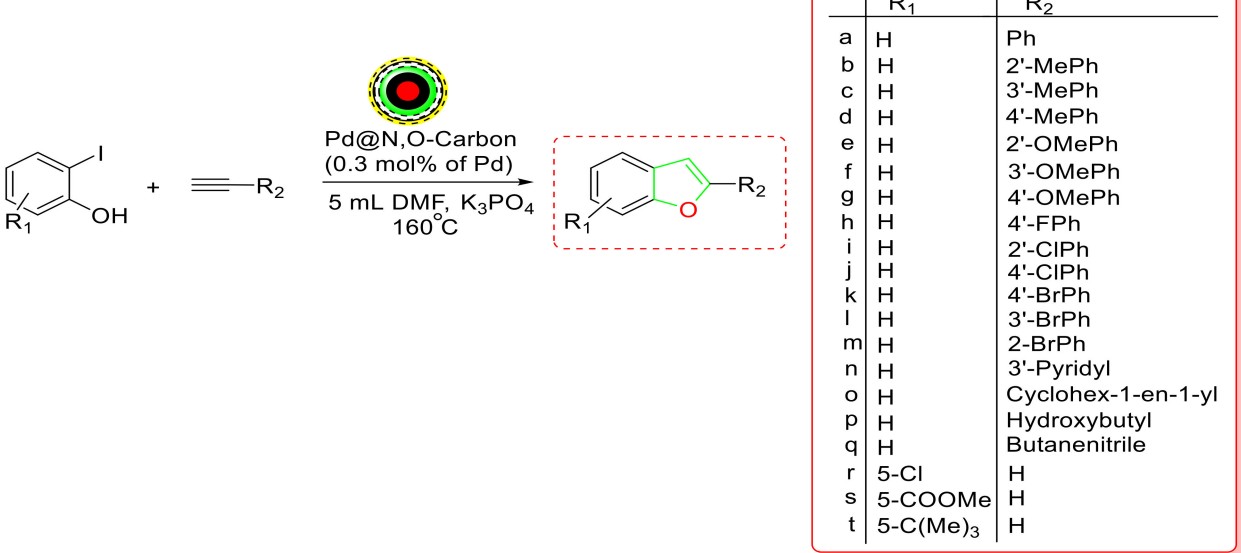

|   | $R_1$ | $R_2$ |
|---|-------|-------|
| a | H | Ph |
| b | H | 2'-MePh |
| c | H | 3'-MePh |
| d | H | 4'-MePh |
| e | H | 2'-OMePh |
| f | H | 3'-OMePh |
| g | H | 4'-OMePh |
| h | H | 4'-FPh |
| i | H | 2'-ClPh |
| j | H | 4'-ClPh |
| k | H | 4'-BrPh |
| l | H | 3'-BrPh |
| m | H | 2-BrPh |
| n | H | 3'-Pyridyl |
| o | H | Cyclohex-1-en-1-yl |
| p | H | Hydroxybutyl |
| q | H | Butanenitrile |
| r | 5-Cl | H |
| s | 5-COOMe | H |
| t | 5-C(Me)$_3$ | H |

**Scheme 1.** Synthesis of 2-benzofurans by using Pd@N,O-Carbon NC.

Using periodic mesoporous organosilica, Baharfar et al. successfully synthesized a novel MgO-containing ionic liquid framework (MgO@PMO-IL) hybrid NC [54]. The produced hybrid composite was applied as a highly efficient and recoverable NC in a one-pot three-component reaction of 1,3-dicarbonyl compounds, N-phenacyl pyridium salts, and isatin to synthesize new spiroindole–furan derivatives (Scheme 2a). The hybrid nanocatalyst utilized could be reused up to seven times without losing efficiency, and it could be easily recovered using a simple filtration method. According to the postulated mechanism (Scheme 2b), the Lewis acidic site ($Mg^{2+}$) activated the carbonyl group of dicarbonyl compounds, causing a nucleophile attack on indole (I). Following that, the activation of pyridium carbonyl formed a nucleophile that attacked the first generated intermediate, I, and finally, the intramolecular cyclization resulted in the formation of matching spiroindole–furan hybrid.

Wang et al. presented an α-cyclodextrin-doped dendritic-fibrous-nanosilica-supported gold NP (DFNS/α-CD/Au NPs) as an effective hybrid NC for efficient carbonylate cinnamyl chloride and phenylacetylene with $CO_2$ to produce 3a,4-dihydronaphtho[2,3-c]-furan-1(3H)-ones [55] (Scheme 3). Because of the lower electron density of alkynes' N-atoms, substrate scope data revealed that alkynes carrying ERG reacted faster, yielding larger yields of furan hybrids. The effectivity of the hybrid DFNS/α-CD/Au NC is due to

the smaller gold particles, which increased the catalytic surface area, and also the higher amount of α-CD.

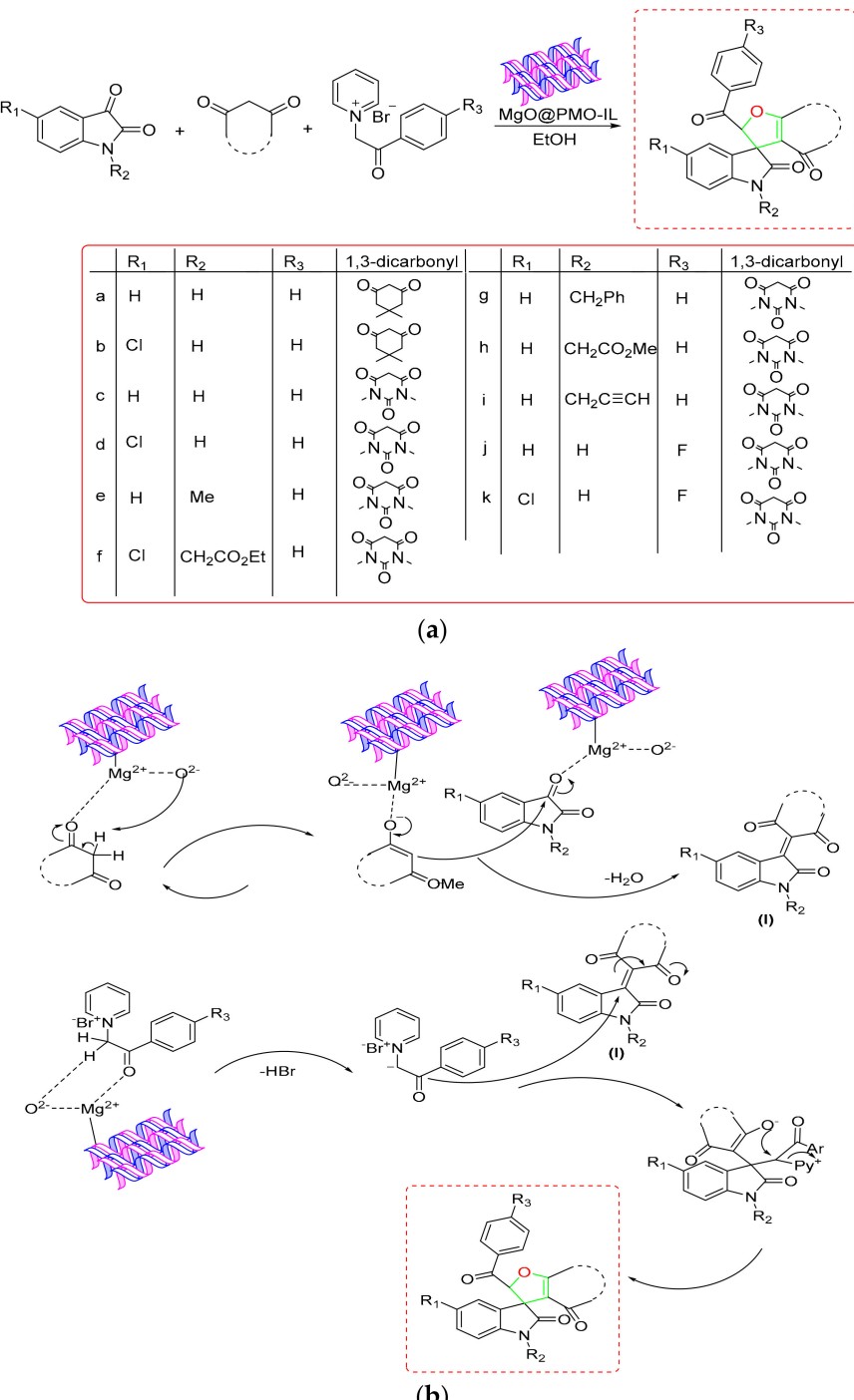

**Scheme 2.** Synthesis of spiroindole–furan derivatives by using MgO@PMO-IL as efficient hybrid NC (**a**); Proposed mechanistic route for the synthesis of spiroindole–furan derivatives by using MgO@PMO-IL hybrid NC (**b**).

Taheri et al. used a unique magnetically recoverable Fe₃O₄@rGO@ZnO-HPA hybrid NC to report an MW-assisted MC green synthesis of Benzo[α]furo[2,3-c]-phenazines [56] (Scheme 4a). According to the mechanistic findings, tautomerism caused the 1,4-dione to produce an intermediate, which then condensed with diamine to form benzo[a]phenazine-5-ol. The aryl-glyoxal and indole condensed at the same time to produce the open shaft. The

intermolecular [3+2] reaction formed the C-C and C-N bonds and finally the loss of water to yield the end product (Scheme 4b). The described technique possesses characteristics such as high product yield, short reaction time, simple methodology, simple workup, recyclable Fe$_3$O$_4$@rGO@ZnO-HPA hybrid NC, and the use of no harmful solvents.

**Scheme 3.** A synthetic protocol for DFNS/α-CD/Au-NC-catalyzed synthesis of 3a,4-dihydronaphtho[2,3-c]-furan-1(3H)-ones.

Furan is a renowned O-heterocycle, and several synthetic approaches have been reported so far for. Here, we collected some recent eco-friendly one-pot, three-component reactions between aromatic aldehyde, alkyne, and aromatic amines under different active metal NCs to synthesize different furan derivatives.

- Khodaei et al. developed 4-carboxybenzyl sulfamic acid functionalized Fe$_3$O$_4$ NPs as a new magnetic five-fold recyclable NC for the rapid one-pot production of furan-2(5H)-ones [57] (Scheme 5a). According to substrate scope data, the proposed MNC displayed outstanding bearing strength against diverse substituents on aldehydes or anilines. According to the postulated mechanism, the acetylic carbonyl was protonated and then attacked by aniline to generate an enamine (I). The protonated aldehyde then interacted with the created enamine, forming an intermediate (II) by rearrangement. After dehydration, the carbonyl group of this newly created intermediate (III) was protonated to cyclized, yielding the desired product (Scheme 5b).

- Shirzaei et al. developed a silica-coated magnetic iron NC doped with thiocarbohydrazide for the quick production of furan derivatives [58] (Scheme 5a). According to a substrate scope investigation, the type of substituents affects the product yield and reaction speed. According to the findings, aromatic aldehydes containing ERG (electron-releasing groups) produced better yields in less time than EWGs (electron-withdrawing groups). The mechanistic research showed that an intermediate (I) was formed via the condensation of aniline and acetylene, which then condensed with activated aldehydes to produce a new intermediate (II). The required derivative of furan was obtained via an intramolecular Michael addition and proton removal of this newly formed intermediate (III) (Scheme 5c). The five-time reusability and high degree of activity make the proposed NC superior to other nonmagnetic catalysts.

- Using sulfamic-acid-2-Aminobenzothiazole-6-carboxylic-acid-adorned Fe$_3$O$_4$ NPs, Hao et al. created a novel organic–inorganic hybrid NC (SA-ABTCA-Fe$_3$O$_4$) that allowed for effective production of 3,4,5-trisubstituted furan-2(5H)-ones [59] (Scheme 5a). The NC was four times recoverable and showed good tolerance for the substituent pattern. According to mechanistic results, 4-aminopyridine and DMAD first created an enamine (I), which then interacted with protonated benzaldehyde to form an intermediate (II). The produced intermediate was subsequently cyclized after undergoing a proton exchange to provide the required product (Scheme 5d). This green NC demonstrated strong reversibility and allowed for a clean operation in a short amount of time, making the protocol more practical and cost-effective.

**Scheme 4.** A systematic presentation of a MC green synthesis of Benzo[$\alpha$]furo[2,3-c]-phenazines by using Fe$_3$O$_4$@rGO@ZnO-HPA as efficient hybrid NC (**a**); Proposed reaction mechanism for the synthesis of Benzo[$\alpha$]furo[2,3-c]-phenazines using Fe$_3$O$_4$@rGO@ZnO-HPA hybrid NC (**b**).

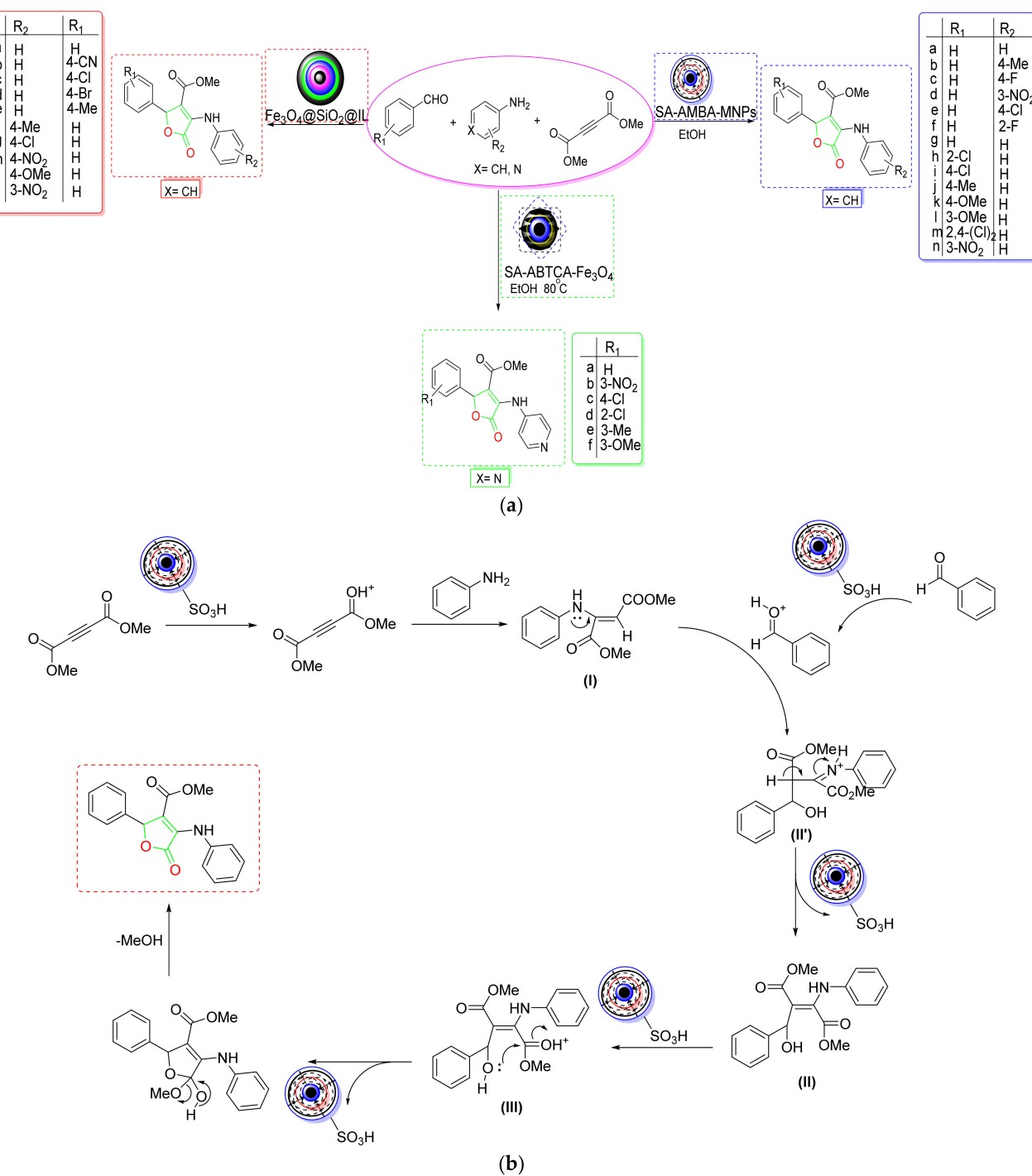

**Scheme 5.** *Cont*.

**Scheme 5.** The schematic representation to design furan derivatives by utilizing different NCs (**a**); Suggested mechanism to synthesize furan-2(5H)-ones by using SA-ABTCA-Fe$_3$O$_4$ (**b**); Reported mechanistic pathway for the nanocatalyzed synthesis of furan derivatives (**c**); Proposed mechanism to achieve 3,4,5-trisubstituted furan-2(5H)-ones by using organic-inorganic hybrids NC (**d**).

### 2.2. Synthesis of Chalcones—The Heterocyclic Intermediate

Though chalcone does not possess a heterocyclic structure, it is structurally related to flavonoids and is used as a common intermediate for synthesizing various heterocyclic flavonoid-type structures and is therefore studied along with flavonoids [60]. Its $\alpha$, $\beta$-unsaturated carbonyl group is responsible for its variety of pharmacological potency and is also used to produce different heterostructures with diverse biological profiles [61–64]. Therefore, chalcone can be considered the core part of different O-heterocyclics. Some of the nanocatalyzed green syntheses of chalcones are as follows:

To combat aggregation, Yadav et al. used the sol-gel process to fabricate Cu@DBM@AS-MNPs, a seven-time-recyclable green and efficient silica-coated magnetically separable hybrid NC [65]. For the solvent-free one-pot $A^3$ coupling to the synthesis chalcone derivatives (Scheme 6), the suggested NC was shown to be impenetrable. Substrate scope data revealed that pyridine-2-carboxaldehyde and its derivatives gave a higher yield of chalcones, but non-aromatic p-substituted aldehydes gave a lower yield of chalcones with a longer reaction time. According to the catalytic protocol, the synthesis was carried out via the isomerization–hydrolysis reaction of aldehyde and terminal alkyne.

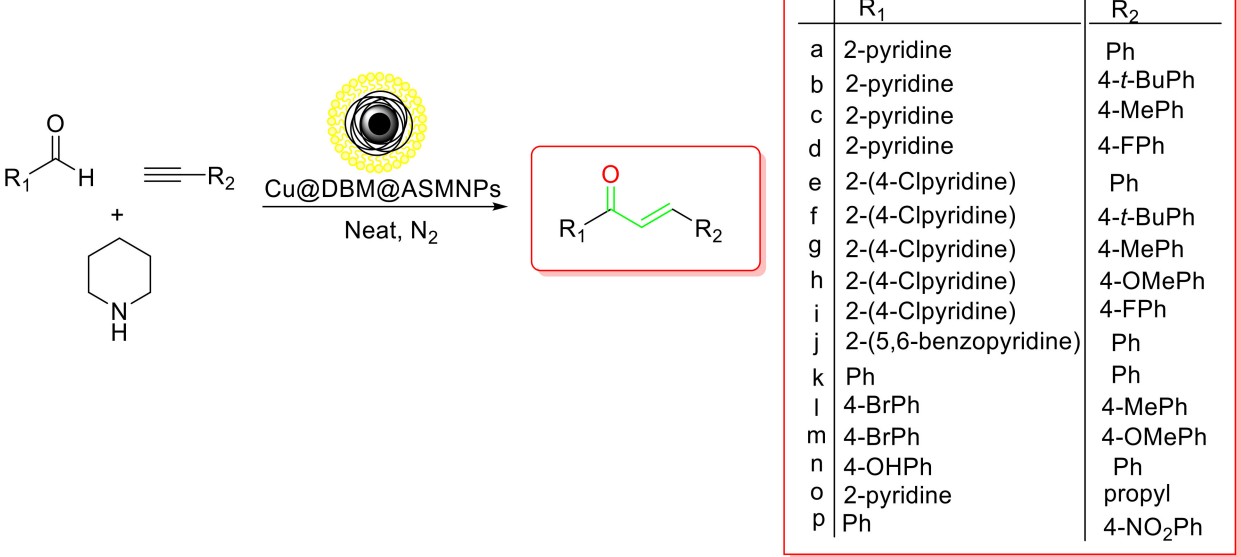

| | $R_1$ | $R_2$ |
|---|---|---|
| a | 2-pyridine | Ph |
| b | 2-pyridine | 4-*t*-BuPh |
| c | 2-pyridine | 4-MePh |
| d | 2-pyridine | 4-FPh |
| e | 2-(4-Clpyridine) | Ph |
| f | 2-(4-Clpyridine) | 4-*t*-BuPh |
| g | 2-(4-Clpyridine) | 4-MePh |
| h | 2-(4-Clpyridine) | 4-OMePh |
| i | 2-(4-Clpyridine) | 4-FPh |
| j | 2-(5,6-benzopyridine) | Ph |
| k | Ph | Ph |
| l | 4-BrPh | 4-MePh |
| m | 4-BrPh | 4-OMePh |
| n | 4-OHPh | Ph |
| o | 2-pyridine | propyl |
| p | Ph | 4-NO$_2$Ph |

**Scheme 6.** The synthetic route to design chalcone via solvent-free one-pot $A^3$ coupling reaction.

Akbarzadeh et al. developed a new mesoporous AI-sba-15 hybrid NC that was modified by N,N′-(1,2-phenylene)bis(2-aminobenzamide) dichloro cobalt and successfully used in manufacturing of 3-cinnamoyl coumarin via a three-component reaction aided by US (Scheme 7a) [66]. According to the hypothesized mechanism, the Co from the organic–inorganic hybrid nanocatalyst (CoCl$_2$-NN′PhBIA/AL-SBA-15) activated the carbonyl group by coordinating with the O-atom. The O-atom from salicylic carbonyl initially formed a compound that interacted with EAA's active methylene group. Finally, transesterification and dehydration were used to produce 3-acetylcoumarin. Following that, O from 3-acetylcoumarin carbonyl reacted with Co from the NC, resulting in aromatic aldehyde aldol condensation (Scheme 7b). The nanocatalyst's N-atom stabilized the enolic hydroxyl and sped up the dehydration process, yielding 3-cinnamoyl coumarins.

The large surface area, Co coordination to speed up nucleophilic addition, and the triggered aldol and dehydration process via H-bonding with the nanocatalyst's N-atom all contribute to make the synthesis more efficient and significant.

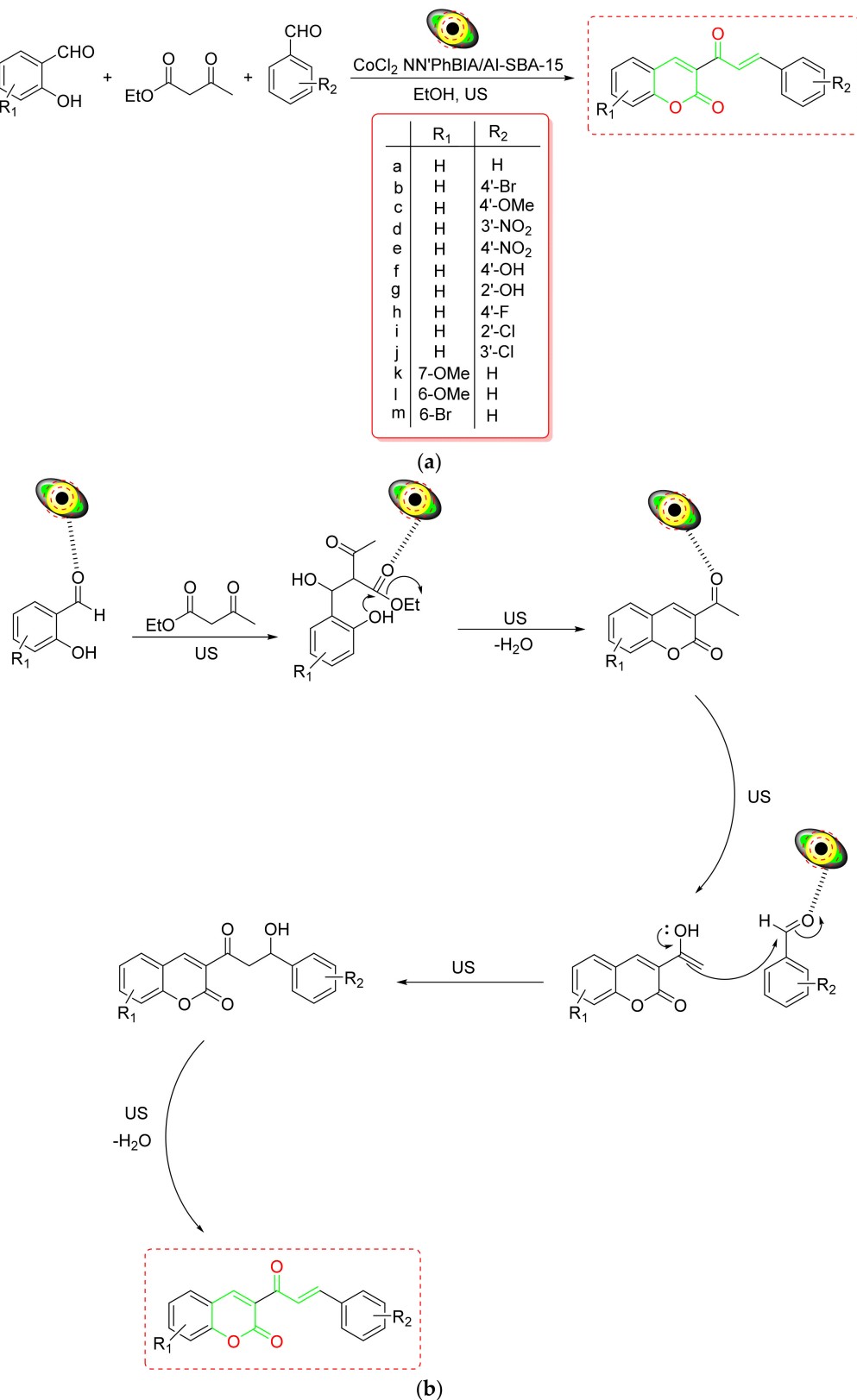

**Scheme 7.** A multicomponent synthesis to design 3-cinnamoyl coumarin by using CoCl$_2$ NN′PhBIA/Al-SBA-15 hybrid NC (**a**); A possible mechanistic pathway to synthesize 3-cinnamoyl coumarin using an organic–inorganic hybrid NC (**b**).

Claisen–Schmidt is a base-catalyzed synthesis of chalcones. Here, we collected some recent Claisen–Schmidt condensations utilizing different NCs following the principles of green chemistry:

- Aryan et al. created a series of unique nanocomposites by employing manganese ferrite (MnFe$_2$O$_4$ NPs) catalyst to modify the surface of natural clinoptilolite [67]. In the aldol-type Claisen–Schmidt process to produce chalcones (Scheme 8a), one of the nanocatalysts demonstrated strong catalytic performance. The proposed synthesis was made faster and more efficient due to robust catalytic synergy between MnFe$_2$O$_4$ NPs and the natural clinoptlolite interface. The substrate scope revealed that the suggested NC was highly tolerant of reactant substituents. According to the mechanistic proof, the acetophenone and benzaldehyde were activated by the nanocatalyst's dual Lewis acidic and Bronsted basic features. The presence of O$^{2-}$ in the NC made the enolate formed from acetophenone more nucleophilic, which initiated the attack on benzaldehyde, resulting in the formation of an intermediate (I) that, following dehydration, supplied the desired product (Scheme 8b). The NC was readily filtered out of the reaction mixture and reused up to four times without losing any catalytic performance.

- Borade and his colleagues employed a sol-gel autoignition approach to produce the zinc ferrite (ZnFe$_2$O$_4$ NPs) which they used as a green fuel and as an efficient NC for manufacturing chalcones (Scheme 8a) [68]. The proposed nanocatalyst's Lewis acidic site aided in the enolization of aryl ketone and activated the benzaldehyde carbonyl group for nucleophilic attack (Scheme 8c). Overall, the proposed approach was shown to be efficient due to its effective MW assistance, solvent-free condition, eco-friendliness, shortened reaction time, fast recovery, and continuous five-cycle reuse of NC.

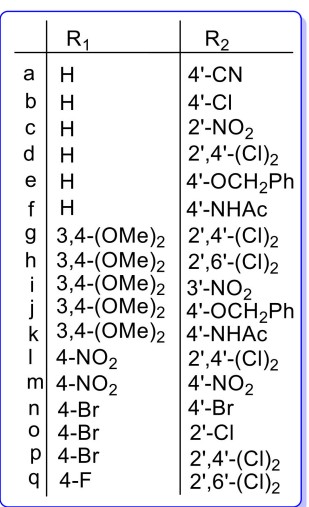

|   | R$_1$ | R$_2$ |
|---|---|---|
| a | H | 4'-CN |
| b | H | 4'-Cl |
| c | H | 2'-NO$_2$ |
| d | H | 2',4'-(Cl)$_2$ |
| e | H | 4'-OCH$_2$Ph |
| f | H | 4'-NHAc |
| g | 3,4-(OMe)$_2$ | 2',4'-(Cl)$_2$ |
| h | 3,4-(OMe)$_2$ | 2',6'-(Cl)$_2$ |
| i | 3,4-(OMe)$_2$ | 3'-NO$_2$ |
| j | 3,4-(OMe)$_2$ | 4'-OCH$_2$Ph |
| k | 3,4-(OMe)$_2$ | 4'-NHAc |
| l | 4-NO$_2$ | 2',4'-(Cl)$_2$ |
| m | 4-NO$_2$ | 4'-NO$_2$ |
| n | 4-Br | 4'-Br |
| o | 4-Br | 2'-Cl |
| p | 4-Br | 2',4'-(Cl)$_2$ |
| q | 4-F | 2',6'-(Cl)$_2$ |

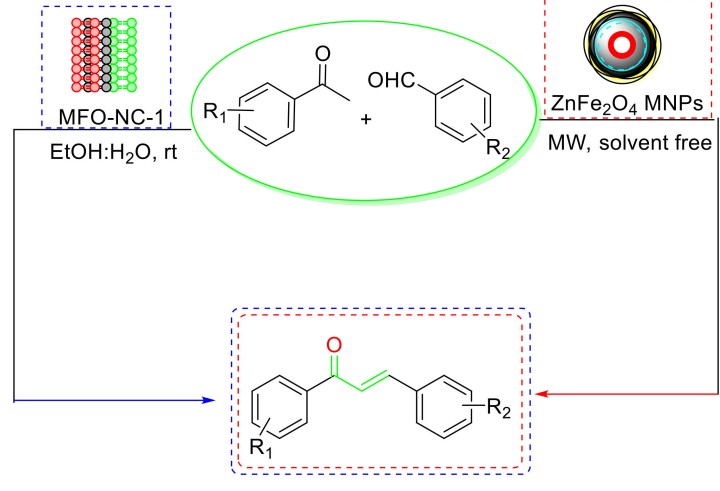

|   | R$_1$ | R$_2$ |
|---|---|---|
| a | H | 4'-CN |
| b | 4-Cl | H |
| c | H | 4'-Cl |
| d | H | 4'-NO$_2$ |
| e | 4-OMe | 4'-NO$_2$ |
| f | 4-Cl | 4'-NMe$_2$ |
| g | 4-OMe | 4'-Cl |
| h | H | 4'-NMe$_2$ |
| i | H | furyl |
| j | 4-OMe | furyl |

(**a**)

**Scheme 8.** *Cont.*

(b)

(c)

**Scheme 8.** A systematic presentation of Claisen–Schmidt condensation by using different NC (**a**); The systematic mechanism for the nanocatalyzed Claisen–Schmidt condensation for chalcones (**b**); The possible mechanism to achieve the synthesis of chalcone derivatives by utilizing $ZnFe_2O_4$ NC (**c**).

### 2.3. Synthesis of Coumarins

Naturally occurring coumarin is an essential class of benzopyrones that act as a structural subunit in many heterocyclic compounds [69] and therefore can be considered as

a promising scaffold to trigger several biological activities [70–73]. Some of the reported nanocatalyzed sustainable syntheses of coumarin derivatives are:

Ghomi and his colleagues introduced $MgFe_2O_4$ NPs as a new NC for the efficient US-assisted Knoevenagel condensation of coumarin derivatives [74] (Scheme 9a). According to the mechanistic analysis, the Mg from the NC and the salicylic carbonyl O-atom produced a complex that reacted with the active methylene group of EAA. Finally, after transesterification and dehydration, the appropriate 3-acetylcoumarin was produced (Scheme 9b). The reported synthetic technique is faster and more efficient due to the increased NP surface area and the cavitation impact of US irradiation.

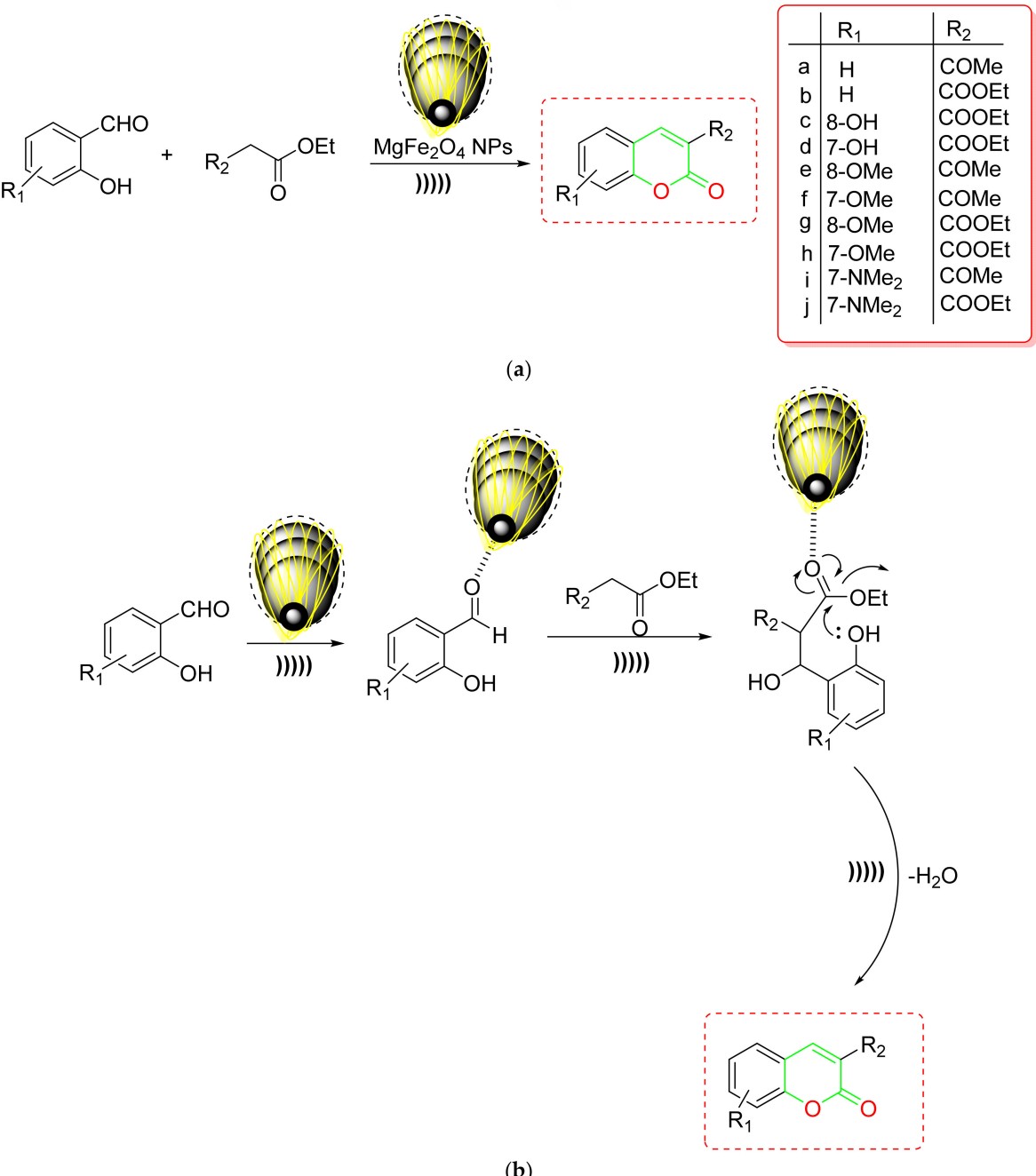

|   | $R_1$ | $R_2$ |
|---|---|---|
| a | H | COMe |
| b | H | COOEt |
| c | 8-OH | COOEt |
| d | 7-OH | COOEt |
| e | 8-OMe | COMe |
| f | 7-OMe | COMe |
| g | 8-OMe | COOEt |
| h | 7-OMe | COOEt |
| i | $7-NMe_2$ | COMe |
| j | $7-NMe_2$ | COOEt |

(**a**)

(**b**)

**Scheme 9.** The synthetic pathway for an US-assisted Knoevenagel condensation of coumarins using $MgFe_2O_4$ NC (**a**); A proposed mechanism for Knoevenagel condensation by using $MgFe_2O_4$ NC (**b**).

Samiei et al. developed a new sulfonated C-coated magnetic (SCCMNPs; $Fe_3O_4$@C@ $OSO_3H$) hybrid NC for efficient Pechmann condensation to produce coumarin derivatives [75] (Scheme 10a). The proposed NC could be reused up to fifteen times without losing catalytic performance. Data from the substrate scope revealed that several ER substituted phenols produced outstanding yields in short reaction times. Catechol and 1-naphthol, on the other hand, required a longer response time due to steric hindrance. Electrophilic aromatic substitution, dehydration, and transesterification were the three steps in the Pechmann synthesis mechanism (Scheme 10b). The proposed mechanism relied heavily on the predicted HOMO–LUMO, NBO atomic charges, and MEP of reactants. Finally, DFT calculation revealed that the reaction was performed through electrophilic attack, dehydration, and finally transesterification, respectively.

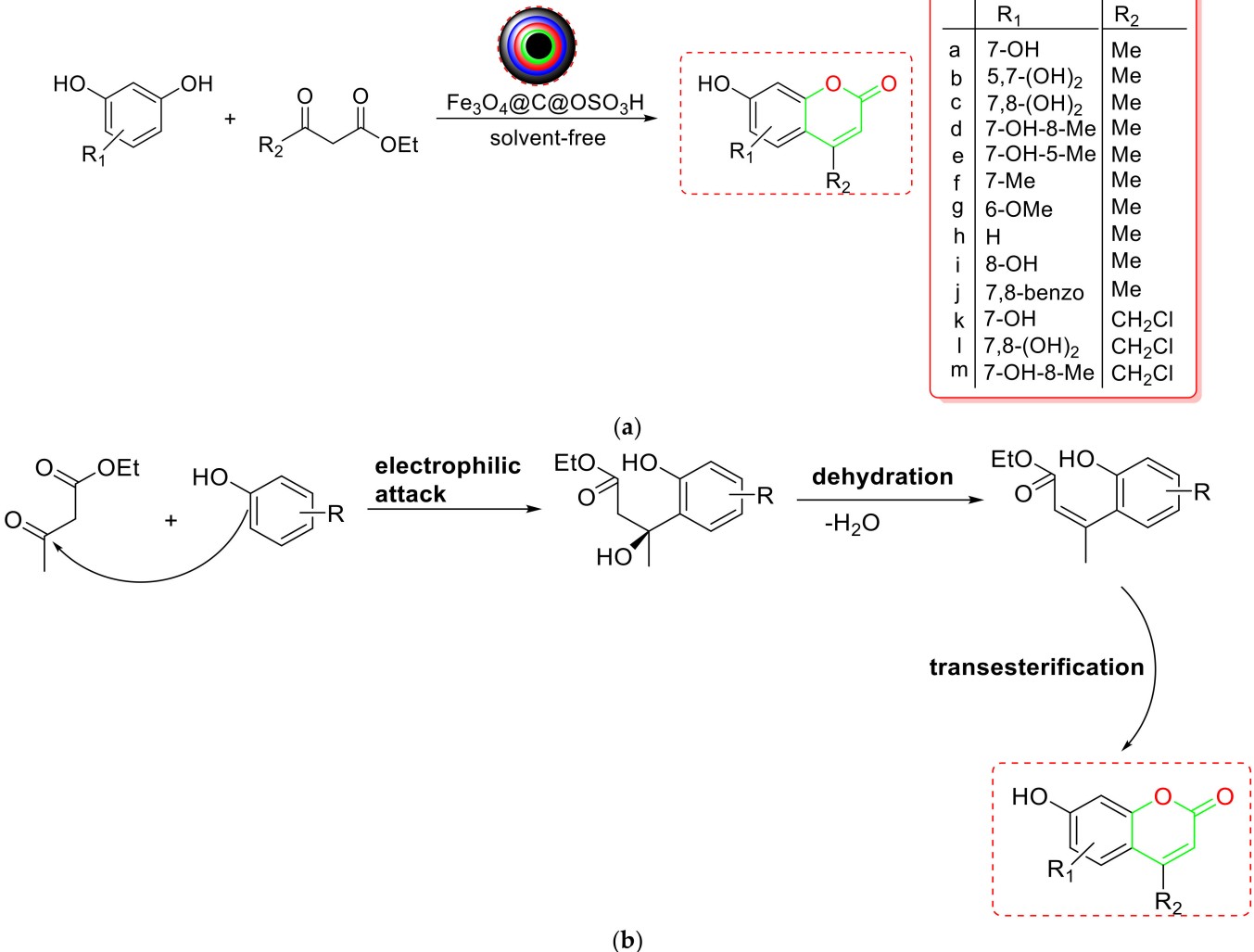

**Scheme 10.** The synthetic route for $Fe_3O_4$@C@$OSO_3H$-nanocatalyzed Pechmann condensation for coumarin (**a**). The plausible mechanistic pathway for the sulfonated C-coated magnetic-nanocatalyzed Pechmann condensation (**b**).

Pechmann condensation is a well-known acid catalyzed synthesis of coumarins, starting from a phenol and a compound containing a β-carbonyl group. Different from the conventional method, here we summarized some recent nanocatalyzed Pechmann condensation under the principles of green chemistry.

- Mirosanloo et al. fabricated a novel biosupported (CNC-AMPD-Pd) NC on palladium NPs utilizing 2-Amino pyrimidine nanocellulose as a support and tested its

catalytic performance in Pechmann condensation to generate coumarin derivatives [76] (Scheme 11a). The proposed NC could be easily recycled and reused up to four times without losing any catalytic activity. According to the proposed mechanism, the CNC-AMPD-Pd catalyzed Pechmann condensation generated an olefinic bond by dehydration while concurrently eliminating ethanol to obtain the required coumarin (Scheme 11b). The proposed technique has the advantages of a reusable catalyst, solvent-free environment, no Pd leaching into the reaction solution, faster reaction times, and simple setup.

- Pakdel et al. proposed a six-time-recyclable magnetic-core-shell-like $Fe_3O_4$@Boehmite-$NH_2$-$Co^{II}$ NPs for solvent-free Pechmann condensation [77] (Scheme 11a). The reliance of suggested NC on the phenol substitution pattern was revealed by the substrate scope discovery. The results showed that phenols with ERG-generated high yields of the intended product, whereas EWG were found to be less reactive or unreactive, as the presence of ERG was actually responsible for the nucleophilic addition of phenol to the carbonyl group of β-ketoester. Mechanistic findings showed that the β-ketoester was first activated and was then attacked by substituted phenols, resulting in the formation of an intermediate (I). Then, after the rearomatization of (I), a new intermediate (II) was formed, of which simultaneous transesterification and ring closure finally produced coumarin after dehydration (Scheme 11c).

- To enable effective US-assisted coumarin production (Scheme 11a), Zarei et al. produced a new magnetic $HFe(SO_4)_2 \cdot 4H_2O$-chitosan nanocomposite [78] ($HFe(SO_4)_2 \cdot 4H_2O$-Ch NCs). According to substrate scope data, phenols containing ERG produced coumarin at a faster rate and with a higher yield than phenols containing EWG. Due to steric hindrance and strong electronegativity, EAA reacted faster than ethyl-4-nitroacetoacetate and 4-chloroacetoacetate. The reported catalyst is easily separated by an external magnetic field, and it can be recycled and reused up to seventeen times. These characteristics make the NCs the most efficient catalyst.

- Yuan et al. developed a copper-supported 5-oxo-4,5-dihydropyrrole-3-carboxylic-acid-functionalized $Fe_3O_4$ NPs (Cu(II)-OHPC-$Fe_3O_4$) as a new magnetic NC for green, solvent-free coumarin synthesis via Pechmann condensation [79] (Scheme 11a). According to the mechanistic findings, the activated EAA first encountered a nucleophilic attack from phenol, forming an intermediate (I) that underwent intramolecular cyclization. After removing the ethanol, the appropriately generated cyclic intermediate (II) yielded the required coumarin (Scheme 11d). The nanocatalyst's large surface area, chemical stability, four-time recyclability, low leaching into the environment, and superior accessibility make it more appealing.

- For the US-assisted green synthesis of coumarin derivatives [80] (Scheme 11a), Bonab et al. developed a new magnetic core-shell of $Fe_3O_4$@c@PeS-$SO_3H$ NC. During the investigation of the substrate scope, it was discovered that the type and position of phenol substituents had a significant impact on the catalyst's performance. In contrast to the EWG, the phenols with ERG promoted synthesis in a shorter reaction time. According to the mechanistic protocol, the carbonyl group of the β-ketoester was first protonated with the proposed NC to render it vulnerable to phenol's nucleophilic assault. Next, electrophilic phenol substitution is followed by dehydration to produce intermediate (I). The NC protonated the generated intermediate (I), which subsequently underwent intermolecular esterification, ethanol elimination, and ring closure to yield the appropriate coumarin (Scheme 11e).

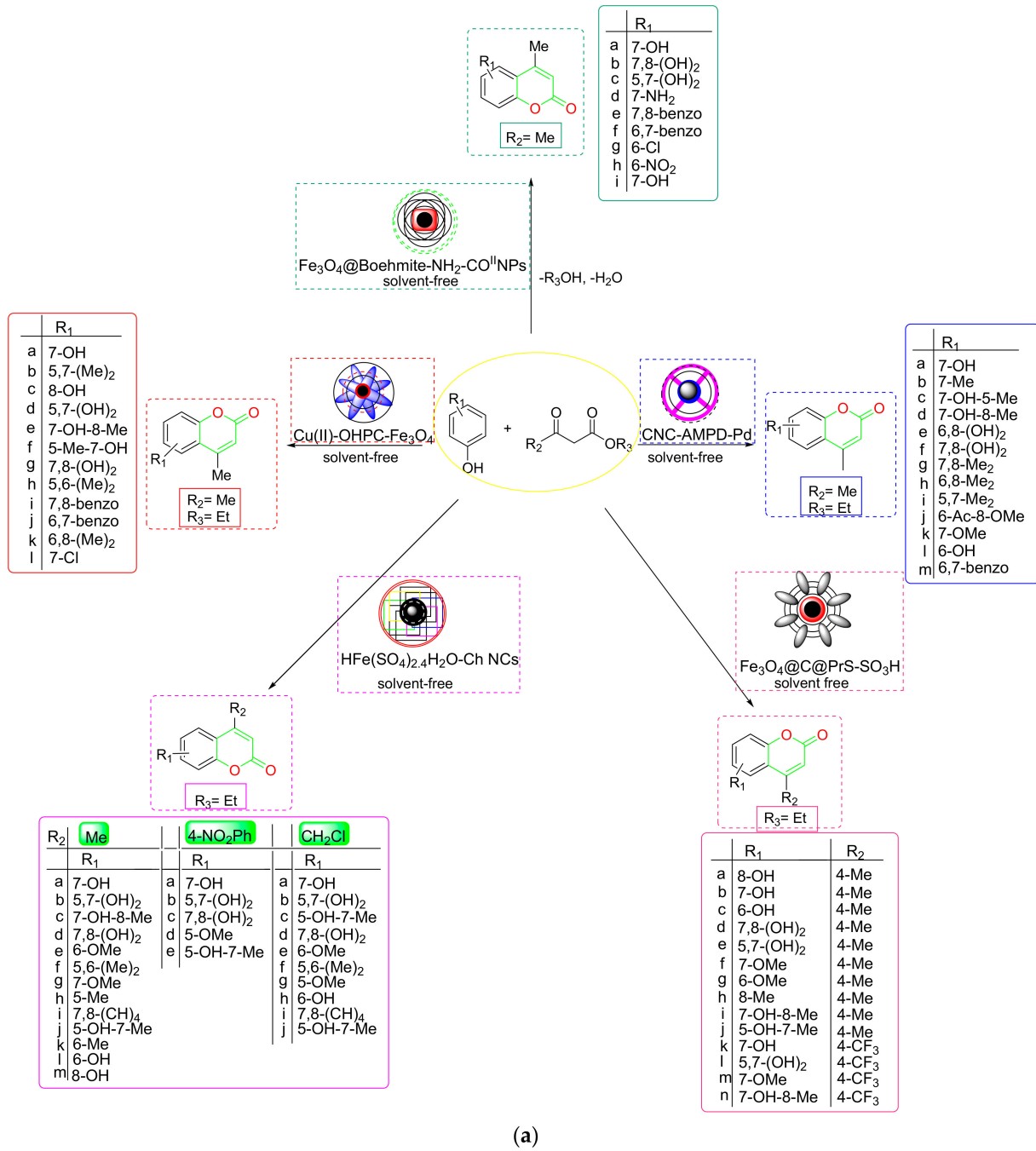

(b)

(c)

**Scheme 11.** *Cont.*

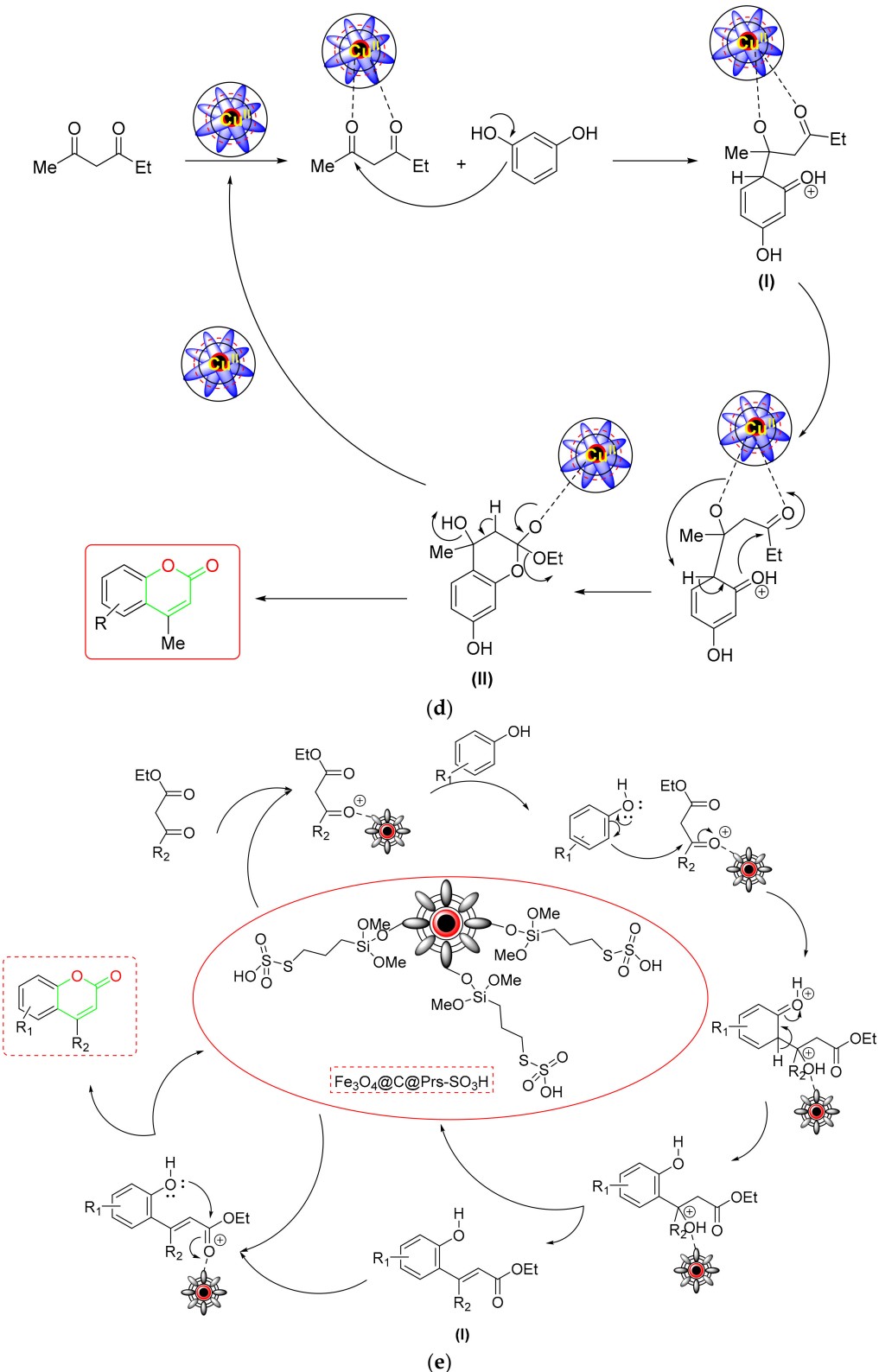

**Scheme 11.** A systematic presentation for the synthetic route for Pechmann condensation by utilizing different NCs (**a**); The systematic mechanistic pathway for the synthesis of coumarins by utilizing 2-Amino pyrimidine nanocellulose as solid support (**b**); A proposed mechanism for the $Fe_3O_4$@Boehmite-$NH_2$-$Co^{II}$-nanocatalyzed Pechmann synthesis for coumarin derivatives (**c**); A possible mechanistic route for Cu(II)-OHPC-$Fe_3O_4$-nanocatalyzed Pechmann condensation (**d**); The mechanistic route to synthesize coumarin derivatives by utilizing a core-shell $Fe_3O_4$@c@PeS-$SO_3H$ NC (**e**).

### 2.4. Synthesis of Oxazole/Isoxazoles

In recent times, oxazole/isoxazole has gained attention due to its promising importance in medicinal chemistry [81,82]. The substitution pattern on the oxazole/isoxazole ring plays a significant role in establishing biological activities such as antimicrobial [83], anticancer [84], antitubercular [85], anti-inflammatory [86], antiobesity [87], antioxidant [88], etc. Some recent green syntheses of oxazole/isoxazole using NCs are listed below:

Using aqueous extract of clover leaves, Abdolmohammadi et al. created a magnetically separable $Fe_3O_4$ NP catalyst in a green biosynthetic approach [89]. The proposed NC was then used to make oxazole and 1H-pyrrole-oxazole hybrids in a green manner (Scheme 12a). According to mechanistic findings, the reaction between acyl chloride and ammonium thiocyante first produced isothiocyante, which then interacted with NaH in the presence of the NC to yield the intermediate (I). After eliminating HBr, one mole of alkyl bromide was reacted with (I) to form a new intermediate (II), which then went through an intramolecular cyclization followed by acid elimination to yield the desired product (Scheme 12b).

Similarly, the reaction of enamine and acetylenic substances triggered the synthesis of an intermediate (IV) by $Fe_3O_4$ NP catalyst and piperidine. The intermediate (IV) was subsequently reacted with another mole of acetylene to form (V), which then reacted with alkylbromide to produce intermediate (VI). The $Fe_3O_4$ NPs catalyst then removed HBr from (VI) to form a new intermediate (VII), which—following oxidation—yielded pyrrole–oxazole hybrids (Scheme 12b).

Fekri et al. developed an amino-glucose-functionalized, silica-coated $NiFe_2O_4$ NP ($NiFe_2O_4$@$SiO_2$@amino glucose) hybrid NC for ecologically friendly solvent-free benzoxazole synthesis [90] (Scheme 13a). The proposed NC was found to be quite tolerable to various substituents in a substrate scope analysis. According to the proposed mechanism, the NC first activated the aldehyde and 1,2-diaminobenzene to form the corresponding imine (I). The NC further activated this iminium species, which went through intramolecular cyclization and eventually led to the final product (Scheme 13b). This used NC can be easily recovered and reutilized for up to five continuous cycles without altering its catalytic activity.

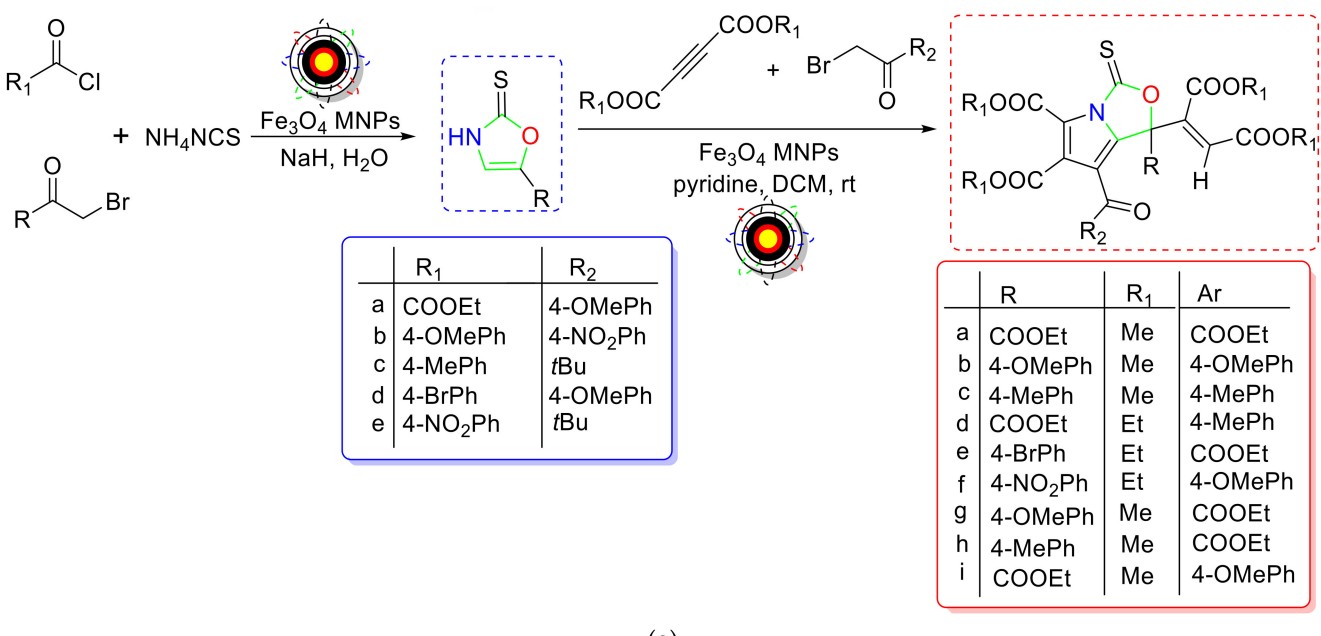

(**a**)

**Scheme 12.** *Cont.*

(**b**)

**Scheme 12.** A systematic reaction pathway to design oxazole and 1H-pyrrole-oxazole hybrids using Fe$_3$O$_4$ NPs catalyst (**a**); The proposed reaction mechanism for Fe$_3$O$_4$-NP-catalyzed synthesis of oxazole and 1H-pyrrole-oxazole hybrids (**b**).

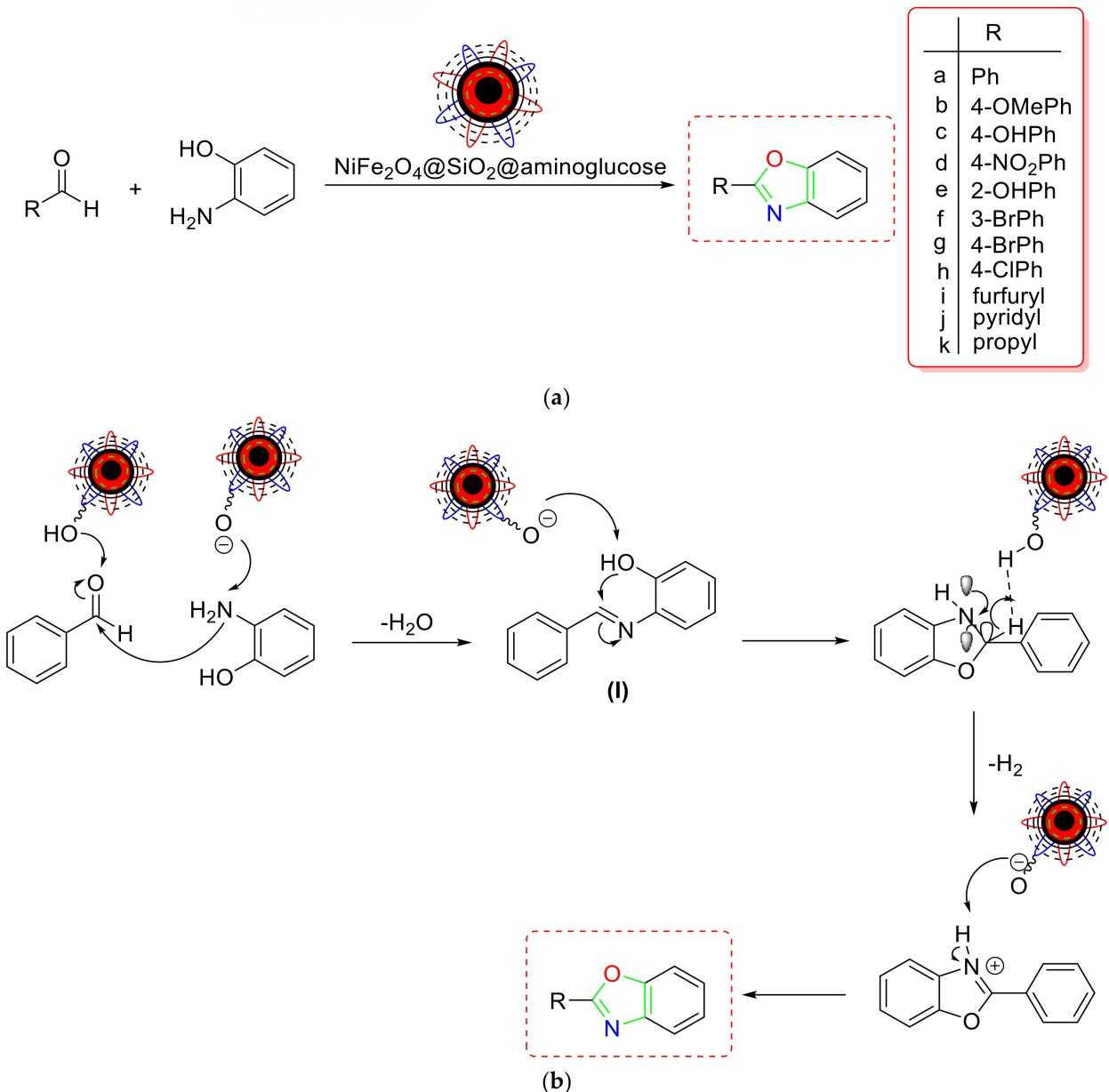

**Scheme 13.** Solvent-free synthesis of benzoxazole by using NiFe$_2$O$_4$@SiO$_2$@aminoglucose NC (**a**); The proposed rection mechanism to synthesize benzoxazoles using NiFe$_2$O$_4$@SiO$_2$@amino glucose NC (**b**).

Aboonajmi et al. successfully proposed an environmentally friendly method for synthesizing benzoxazole heterocycles via the oxidation of catechols followed by condensation, cyclization, and aromatization [91] (Scheme 14a). To access the desired synthesis, an iron(III)-tetraphenylporphyrinato-chloride (FeTPPCl) nanostructure was produced and then utilized for the synthesis. The designed nanostructure was found to be more selective than Sn(II)TPP, Cu(II)TPP, NiTPP, Pb(II)TPP, and CdTPP for this synthesis. Substrate scope investigation suggested that benzylamine with EWG and ERG at the o- and p-positions worked excellently to produce corresponding benzoxazole. Furthermore, aliphatic amines without an aromatic ring also performed well in the chemical sequence to produce benzoxazole. Mechanistic study (Scheme 14b) showed that initially, the catechol was oxidized using FeTPPCl-Np, which then went through a nucleophilic addition with 1°-amine and formed an intermediate. The formed intermediate was dehydrated to produce imine and formed a Schiff base via tautomerization. The Schiff base was then transformed to benzoxa-

zoline via addition–cyclization and finally oxidized to benzoxazole via aerobic oxidative dehydrogenation.

Yu et al. reported a nanocomposite of AgPd NP anchored on WO$_{2.72}$ NRs (AgPd/WO$_{2.72}$) to achieve a one-pot synthesis of benzoxazoles [92] (Scheme 15). Different compositions with various inorganic supports, such as carbon (C), silica (SiO$_2$), or WO$_{2.9}$ on AgPd catalysts were studied. However, the strong interfacial interaction between AgPd and WO$_{2.72}$ was optimal and resulted in the expansion of the AgPd lattice and electron polarization from AgPd to WO$_{2.72}$. This induced polarization led to an increase in the Lewis acidity of AgPd and the Lewis basicity of WO$_{2.72}$ and therefore led to the high activity of the nanocatalyst. The substrate scope data showed that any substituent at the p- and m-positions could be bearable, whereas those at the o-position led to a decrease in the yield due to steric hindrance. Furthermore, the replacement of aromatic aldehyde with aliphatic led to a lower yield.

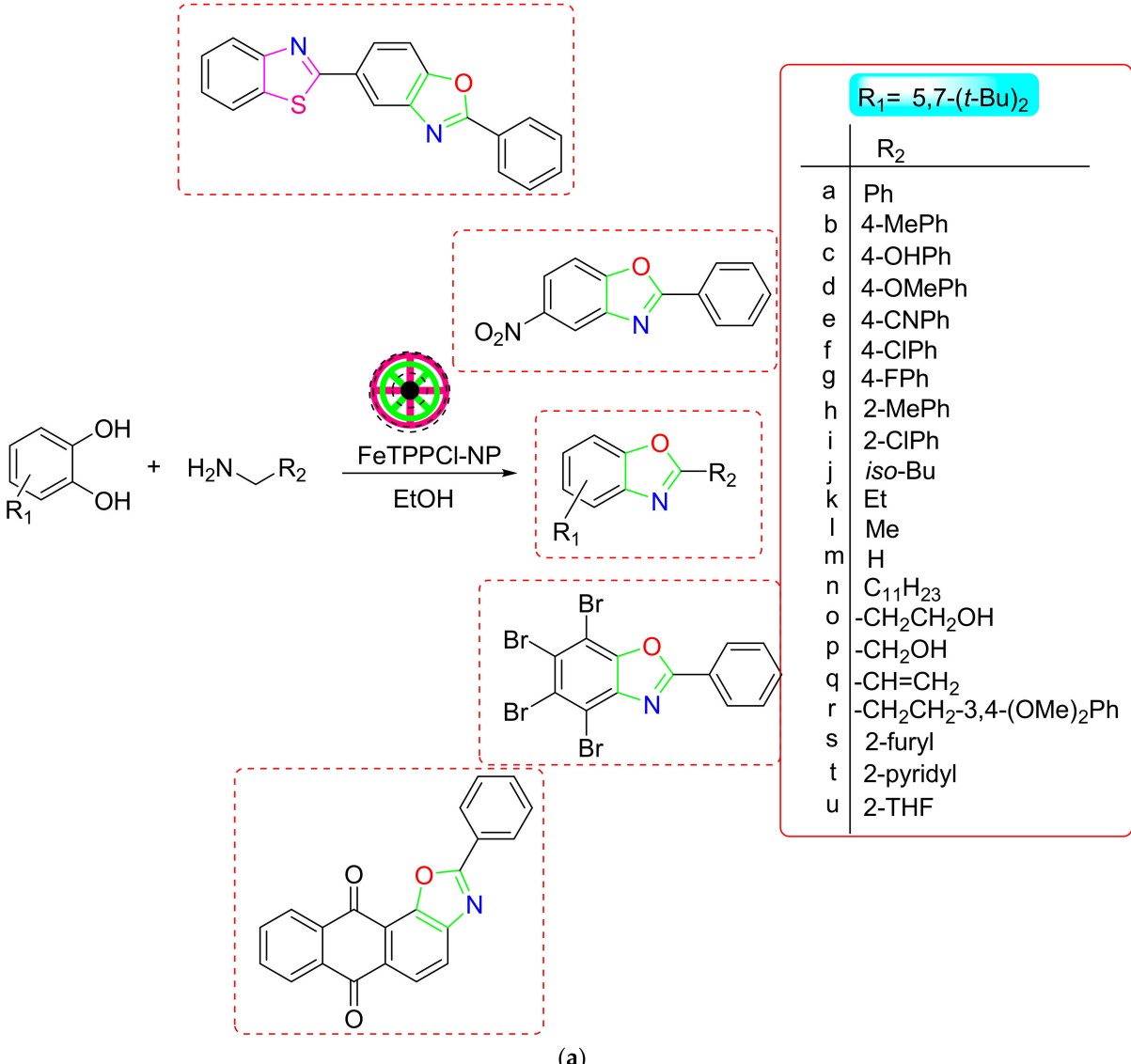

| | R$_1$= 5,7-($t$-Bu)$_2$ |
| --- | --- |
| | R$_2$ |
| a | Ph |
| b | 4-MePh |
| c | 4-OHPh |
| d | 4-OMePh |
| e | 4-CNPh |
| f | 4-ClPh |
| g | 4-FPh |
| h | 2-MePh |
| i | 2-ClPh |
| j | *iso*-Bu |
| k | Et |
| l | Me |
| m | H |
| n | C$_{11}$H$_{23}$ |
| o | -CH$_2$CH$_2$OH |
| p | -CH$_2$OH |
| q | -CH=CH$_2$ |
| r | -CH$_2$CH$_2$-3,4-(OMe)$_2$Ph |
| s | 2-furyl |
| t | 2-pyridyl |
| u | 2-THF |

(**a**)

**Scheme 14.** *Cont.*

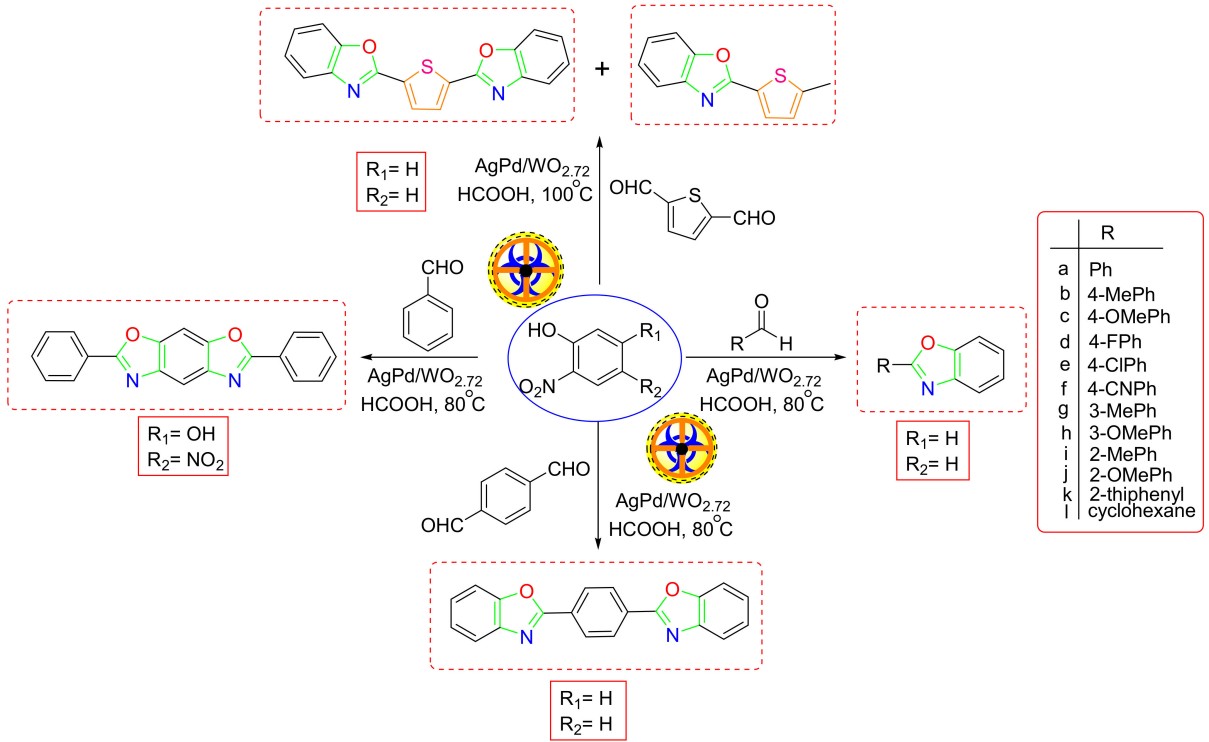

**Scheme 14.** Synthesis protocol for benzoxazole heterocycles using iron(III)porphyrin nanostructure (**a**); A proposed mechanism to synthesize benzoxazole derivatives catalyzed by FeTPPCl Np (**b**).

**Scheme 15.** Synthesis of benzoxazoles by using AgPd/WO$_{2.72}$ NC.

Shanshak et al. created a novel six-time-reusable core-shell NC, ZnO@Fe$_3$O$_4$, to design an effective and environmentally friendly protocol for the synthesis of isoxazole-5(4H)-ones [93] (Scheme 16a). According to substrate scope data, the proposed NC showed a high tolerance power to the substitution pattern on aldehydes. It was discovered that aldehyde with ERG at the p- and m-positions gave a high yield with a quick reaction time; however, p-nitrobenzaldehyde gave the lowest-percentage yield formation product. Mechanistic results showed that the aqueous-medium-assisted MC synthesis began with the condensation of nanocatalyzed activated EAA and hydroxyl amine to produce an intermediate (I), which then underwent Knoevenagel condensation with aldehydes to produce a new intermediate (II). The desired isoxazole derivatives were then obtained by dehydration and intramolecular cyclization of the newly produced intermediate (Scheme 16b).

*2.5. Synthesis of Pyrans*

Over the last several decades, pyran has been considered one of the most prominent members of the O-heterocycles due to its broad range of biological activities [94], such as antimicrobial [95], anticancer [96], anti-HIV [97], anti-inflammatory [98], and antituberculosis [99] activities. Some of the latest reported nanocatalyzed green syntheses of pyran derivatives are:

In the presence of KF/clinoptilolite NCs, Ghotbabadi and coworkers reported a green one-pot aldol condensed/intramolecular cyclization to create 1-[6-hydroxy-2(prop-1-en-2yl)-1-benzofuran-5-yl]ehtanone [100] (Scheme 17a). Because of the H-bonding interaction with the proton of methyl ketone, the reaction began with the creation of an anionic intermediate (I). After aldol condensation between I and aldehyde, a newly generated intermediate (II) underwent dehydration, resulting in the formation of chalcone. The chalcone double bond was then generated as an epoxide in the presence of H$_2$O$_2$, and the final product was obtained via exo-trig cyclization and simultaneous oxidation (Scheme 17b). The technique was successful since it used water as a solvent, had a low supported catalyst loading, and produced little waste.

Ghasemzadeh et al. mounted Zn$^{II}$ on epibromohydrin-functionalized $\gamma$-Fe$_2$O$_3$@SiO$_2$ NPs to construct a new hybrid NC ($\gamma$-Fe$_2$O$_3$@SiO$_2$-EC-Zn$^{II}$) that allows for the green and quick synthesis of spiro[indoline-3,9-xanthene] trione derivatives in aqueous medium [101] (Scheme 18a). According to the substrate scope analysis, dimedone reacted faster than 1,3-cyclohexaedione because dimedone's active methylene is more reactive than 1,3-cyclohexaedione's. The hypothesized mechanism also revealed that the Lewis acidity of the NC was required for the condensation reaction to proceed. Isatin was activated by the Lewis acidic site (Zn$^{II}$) against the nucleophilic assault of an activated enolized 1,3-dicarbonyl group, resulting in an intermediate (I) that dehydrated to provide the aldol adduct. After dehydration, the next step was to start with Michael addition to produce the intermediate (II), which then went through intramolecular cyclization to produce the desired product (Scheme 18b). The NC utilized in the reaction was then recovered and reused for up to seven runs. As a result, the proposed protocol appears to be quite promising, as it provides a better scope for spiroxanthenes synthesis with ecological benefits.

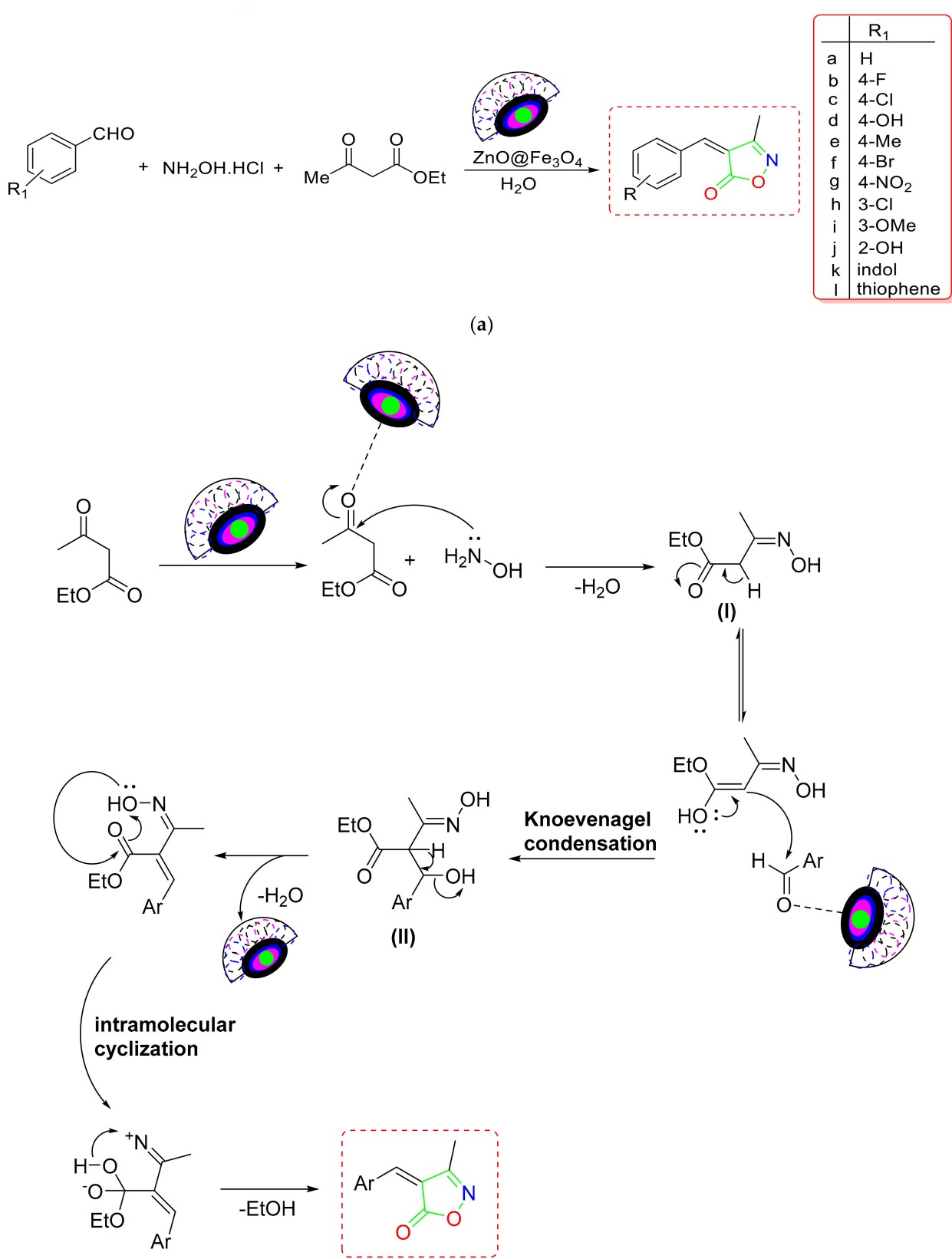

**Scheme 16.** Synthesis of isoxazole-5(4H)-ones by utilizing a core shell ZnO@Fe$_3$O$_4$ NC (**a**); The reported mechanism for the ZnO@Fe$_3$O$_4$-nanocatalyzed synthesis of isoxazole-5(4H)-ones (**b**).

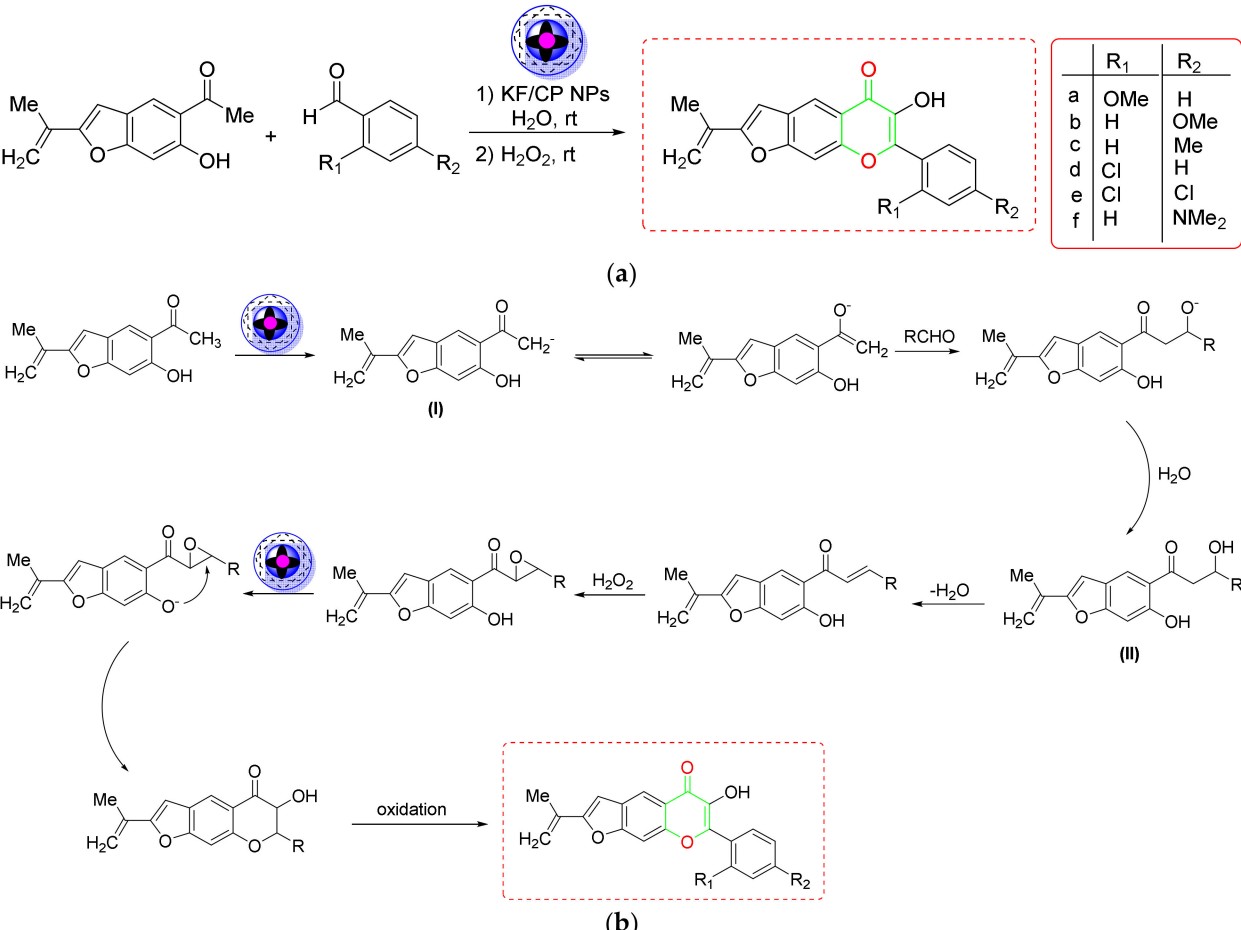

**Scheme 17.** The synthetic pathway to design 1-[6-hydroxy-2(prop-1-en-2yl)-1-benzofuran-5-yl]ehtanone using KF/clinoptilolite NCs (**a**); The proposed reaction mechanism for the synthesis of 1-[6-hydroxy-2(prop-1-en-2yl)-1-benzofuran-5-yl]ehtanone (**b**).

By grafting γ-aminobutyric acid on superparamagnetic γ-Fe$_2$O$_3$@SiO$_2$ hybrid NC, Mohammadi et al. created a novel five-fold-recyclable NC, γ-Fe$_2$O$_3$@SiO$_2$-aminobutyric acid, which they tested for successful green cascade synthesis of chromeno[4,3,2-de][1,6]na-phthyridine derivatives [102] (Scheme 19a). The reaction proceeded efficiently with a wide range of substituents on benzaldehyde, according to the substrate scope analysis. According to the hypothesized mechanism, the reaction began with the synthesis of chalcone by aldol condensation between benzaldehyde and acetophenone, followed by Knoeve-nagel condensation between chalcone and malononitrile. The intramolecular nucleophilic assault of enolic OH on the Knoevenagel product would then result in the creation of an intermediate (I). The recently generated intermediate (I) was then condensed with a second molecule of malononitrile to form a new intermediate (II), which was then followed by intramolecular cyclization and simultaneous aromatization to yield the final product (Scheme 19b).

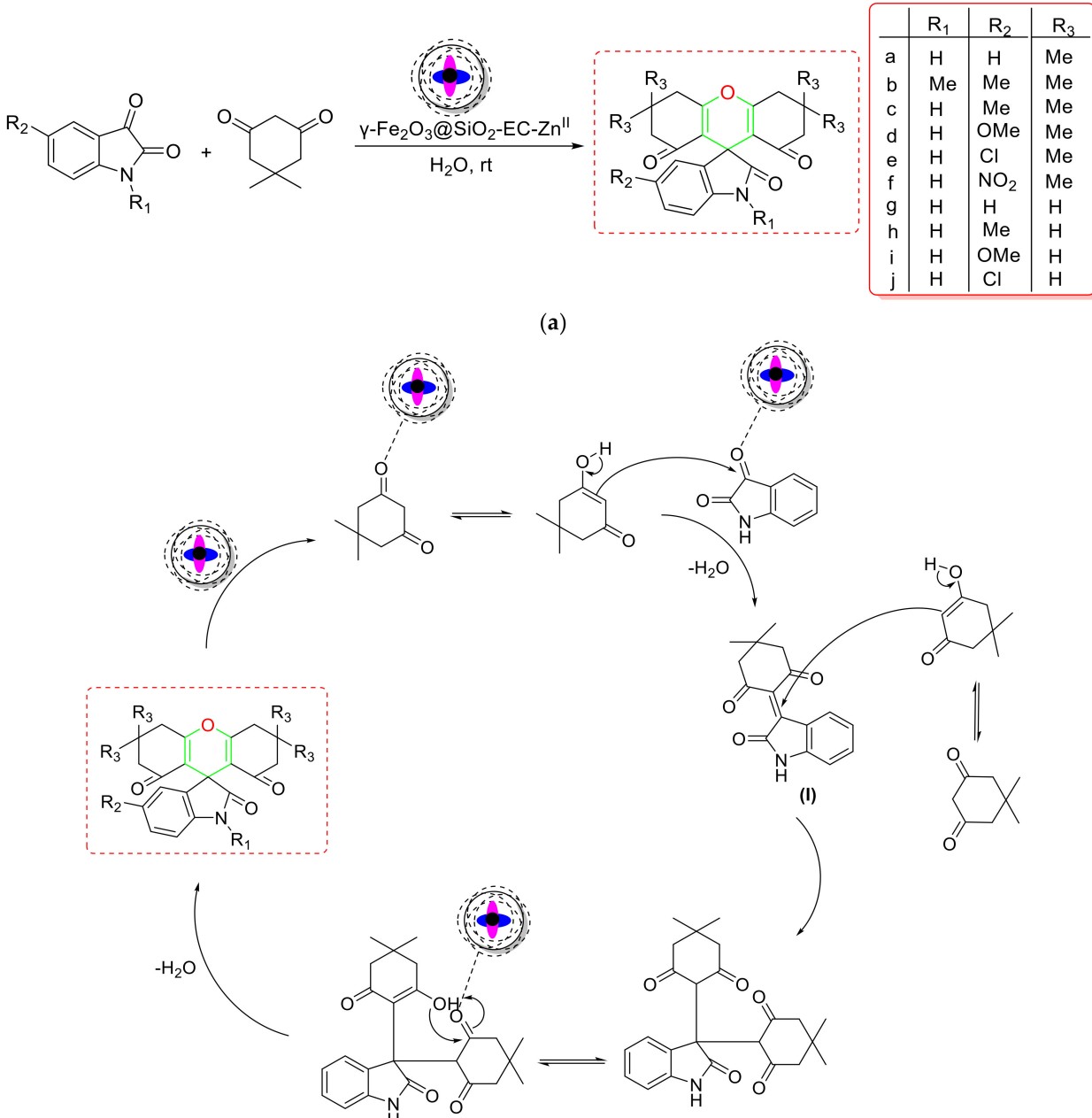

| | R₁ | R₂ | R₃ |
|---|---|---|---|
| a | H | H | Me |
| b | Me | Me | Me |
| c | H | Me | Me |
| d | H | OMe | Me |
| e | H | Cl | Me |
| f | H | NO₂ | Me |
| g | H | H | H |
| h | H | Me | H |
| i | H | OMe | H |
| j | H | Cl | H |

(**a**)

(**b**)

**Scheme 18.** The synthetic pathway for γ-Fe₂O₃@SiO₂-EC-Zn^II-catalyzed spiro[indoline-3,9-xanthene] trione synthesis (**a**); The proposed reaction mechanism for the synthesis of spiro[indoline-3,9-xanthene] triones by using γ-Fe₂O₃@SiO₂-EC-Zn^II hybrid NC (**b**).

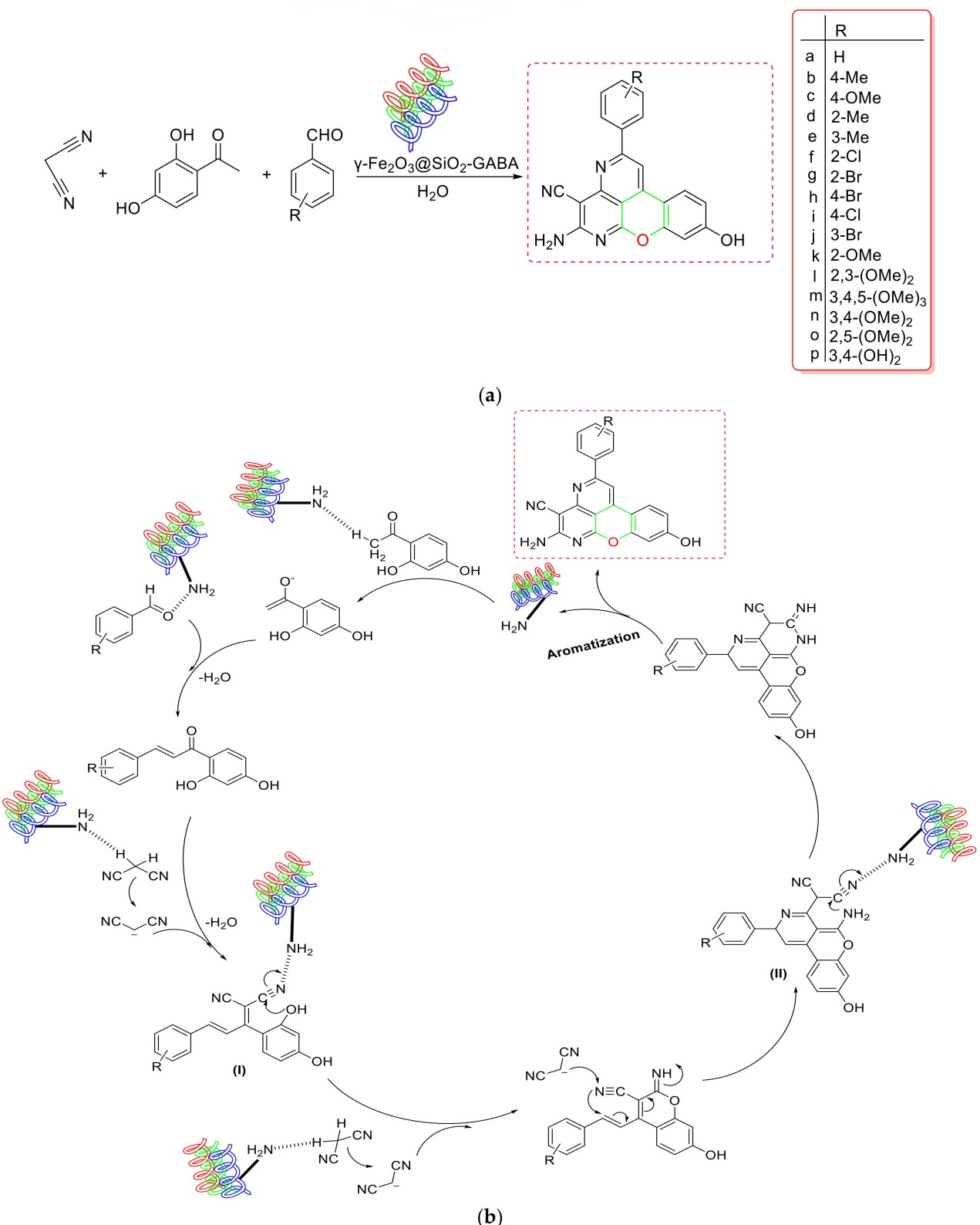

**Scheme 19.** Synthesis of chromeno[4,3,2-de][1,6]naphthyridine using γ-Fe₂O₃@SiO₂-aminobutyric acid hybrid NC (**a**); The plausible mechanism for the γ-Fe₂O₃@SiO₂-aminobutyric-acid-nanocatalyzed synthesis of chromeno[4,3,2-de][1,6]naphthyridine (**b**).

The multicomponent (MC) approach is a powerful tool in drug discovery and combinational chemistry with several advantages, including atom economy and step efficiency. Here, we displayed some MC reactions which afford pyran synthesis under different NCs.

- Eshtehardian et al. used a green and simple method to produce $MgFe_2O_4$ NC, which they tested for the synthesis of 2-amino-7-hydroxy-4H-chromene and tetrahydrobenzo[b]pyran derivatives [103] (Scheme 20a). The proposed NC could be magnetically separated and reused four times without losing its catalytic activity. Based on mechanistic research, it was rationally assumed that Knoevenagel condensation of aldehyde and malononitrile formed an intermediate (I), which then proceeded through Michael addition with resorcinol to produce the adduct (II). To produce the corresponding product, this adduct might have undergone intramolecular cyclization followed by [1,3] H-shift (Scheme 20b). As a result, $MgFe_2O_4$'s Lewis acidic feature enabled Knoevenagel condensation and Michael addition by interacting with the aldehydic carbonyl O-atom and the cyanide group, respectively, and allowed the reaction to proceed successfully. As a result, the Lewis acidic feature of $MgFe_2O_4$ NC enhanced the Knoevenagel condensation and Michael addition by interacting with the aldehydic carbonyl O-atom and the cyanide group, respectively, and allowed the reaction to take place over its large surface area.

- Using a six-time-recoverable nickel ferrite $NiFe_2O_4$ NC, Pourshojaei et al. developed a one-pot cascade synthesis of 4H-chromenes [104] (Scheme 20a). The nanocatalyst's amphoteric Lewis feature makes it useful for the fast synthesis of 4H-chromenes. According to a substrate scope investigation, the type of substituents on benzaldehyde had a significant impact on the effectiveness of the NC. The catalytic activity of the EWG was found to be higher than that of the ERG. According to the mechanistic findings, the Knoevenagel condensation between activated aldehydes and malononitrile first formed an intermediate (I), which then interacted with the activated dimedone and produced a new intermediate (II) after losing a water molecule. The newly created intermediate (II) was then subjected to intramolecular cyclization before being converted to the desired product via imine–enamine tautomerism (Scheme 20c).

- Singh et al. developed a magnetically retrievable amine-decorated $SiO_2@Fe_3O_4$ hybrid NC ($NH_2@SiO_2@Fe_3O_4$) to carry out a solvent-free multicomponent synthesis of 2-amino-4H-benzo[b]pyran derivatives [105] (Scheme 20a). According to the data from the substrate scope study, the designed NC was tolerant of a wide variety of functional groups. According to mechanistic results, the proposed multicomponent method was initiated by the basic amino sites of the NC. The reaction began with the Knoevenagel condensation of malononitrile and aldehyde to produce arylidiene malononitrile, which was then Michael added to dimedone to produce an intermediate (I). To produce the intended product, this produced intermediate (I) was cyclized intramolecularly and then protonated (Scheme 20d). The hybrid NC has numerous notable characteristics, including durability, reusability, and recyclability for up to three reaction cycles, as well as a short reaction time and milder reaction conditions.

Lati et al. presented $Fe_3O_4@SiO_2$-Sultone as a new magnetic hybrid NC for the fast, green, and environmentally friendly synthesis of 3,4-dihydropyrano[c]chromenes and 2-amino-4H-chromenes [106] (Scheme 21a). Substrate scope analysis revealed that the reaction was effective with a variety of substituents. Mechanistic findings suggested that the five-times-recoverable NC catalyst first activated the aldehyde, which was subsequently attacked by activated malononitrile to produce a Knoevenagel-condensed intermediate (I). To produce a new intermediate (II), a Michael addition was performed between the Knoevenagel intermediate (I) and naphthol to form the intermediate (III). Using simultaneous cyclization and tautomerization, this newly created intermediate (III) was then tautomerized to produce the desired product (Scheme 21b).

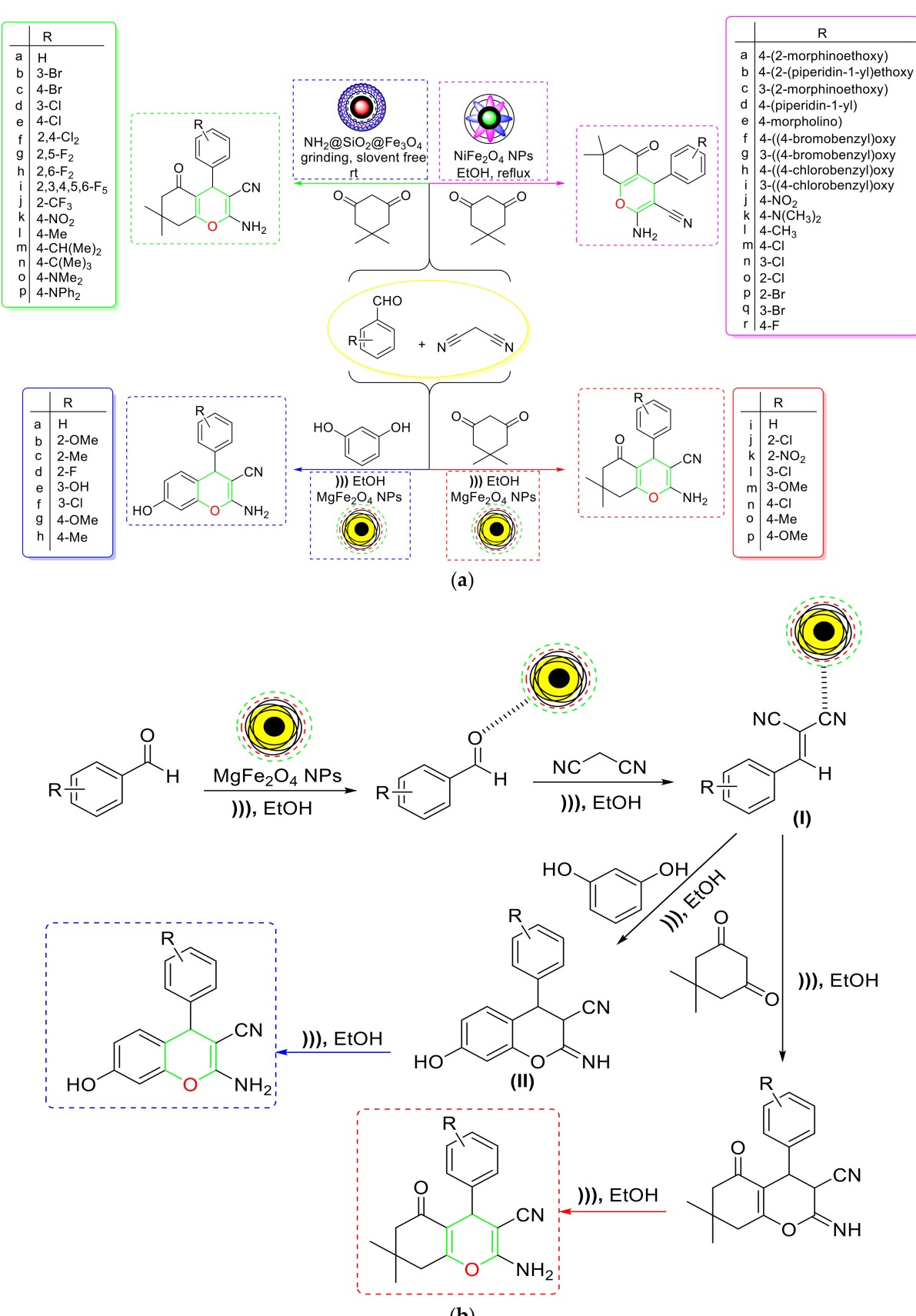

**Scheme 20.** *Cont.*

**Scheme 20.** A systematic protocol to design different pyran derivatives under different reaction conditions (**a**); Mechanistic pathway to synthesize 2-amino-7-hydroxy-4H-chromene/tetrahydrobenzo[b]pyran derivatives by utilizing MgFe$_2$O$_4$ NC (**b**); Possible mechanism for NiFe$_2$O$_4$-NC-catalyzed synthesis of 4H-chromenes (**c**); Mechanistic route to achieve the synthesis of 2-amino-4H-benzo[b]pyrans using an amine-decorated SiO$_2$@Fe$_3$O$_4$ NC (**d**).

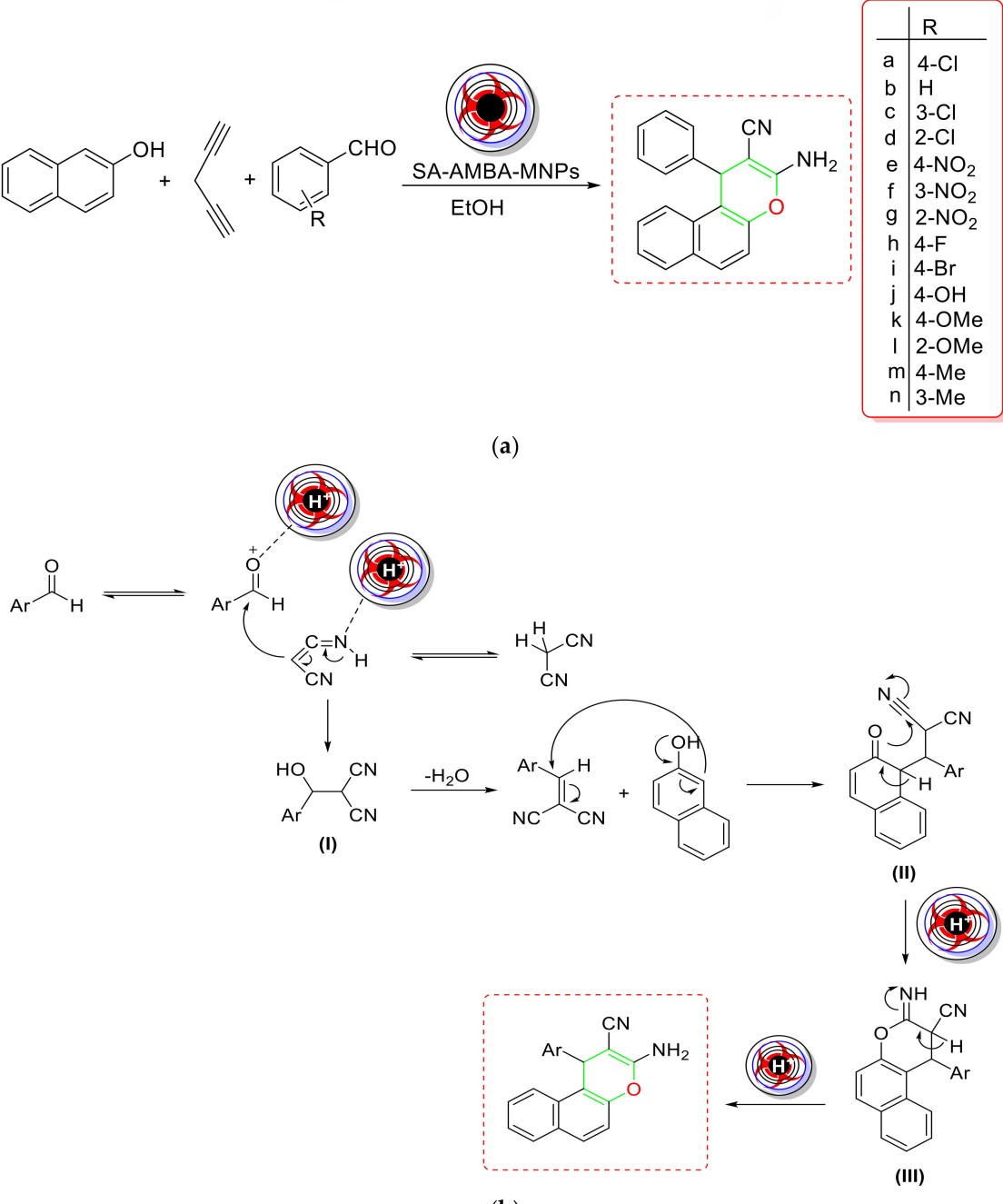

**Scheme 21.** An eco-friendly synthesis of 3,4-dihydropyrano[c]chromenes and 2-amino-4H-chromenes by using SA-AMBA-MNPs NC (**a**); Proposed mechanistic pathway for the synthesis for chromene synthesis catalyzed by $Fe_3O_4$@$SiO_2$-Sultone hybrid NC (**b**).

Knoevenagel condensation, or simply the modification of the aldol condensation reaction, is an organic reaction named after Emil Knoevenagel. It is a well-known condensation which achieves the synthesis of α,β-unsaturated compounds. Here, we reported some green Knoevenagel condensations which synthesize chalcone derivatives by utilizing different NCs.

- Karami et al. developed a magnetic bifunctional ($Fe_3O_4$@$SiO_2$@PTS-DABA) hybrid NC that was found to be catalytically effective for the synthesis of dihydropyranopyrazole [107] (Scheme 22a). According to the mechanistic findings, a pyrazolone ring was first generated by nanocatalyzed condensation between activated EAA and hydrazine

hydrate, followed by dehydration and tautomerization. The Knoevenagel condensation of aldehyde and malononitrile created an intermediate (I), which interacted with pyrazolone to form the desired product by 6-exo-dig cyclization and tautomerization. It was discovered that the catalytic surface's acidic and basic sites promoted intramolecular electrophilic cyclization via tautomerization (Scheme 22b). The features of the proposed process, such as quick synthesis, environmentally acceptable solvent, a five-time-recyclable catalyst, and the avoidance of caustic reagents, all make this technique appealing.

- In line with sustainable chemistry, Kamalzare et al. created a new magnetic biocomposite with chitosan and tannic acid (Fe$_3$O$_4$@chitosan-tannic acid) and used it in a cascade pyranopyrazole synthesis [108] (Scheme 22a). The observed hybrid NC had a high tolerance power for the benzaldehyde substituent pattern. According to one hypothesized mechanism, the proposed hybrid NC activated the carbonyl group of EAA and went through a nucleophilic assault of hydrazine to generate an intermediate, which was then subsequently attacked by another hydrazine molecule and eventually formed the pyrazolone ring (i) after losing water. The Knoevenagel condensation of aldehydes and malononitrile produced a novel intermediate (II), which was subsequently reacted with enolic pyrazolone to produce the desired product after significant intramolecular cyclization and tautomerization (Scheme 22c).

Hoseinzade et al. designed Agar-coated Fe$_3$O$_4$ NPs (Fe$_3$O$_4$@agar) by coprecipitating Fe$^{2+}$ and Fe$^{3+}$ ions in aqueous Agar with NH$_4$OH and then coating them with Ag$^+$ ions, which were then mildly reduced with NBH to give Fe$_3$O$_4$@Agar-Ag NPs. The suggested hybrid NC was then tested for its catalytic activity in the fast synthesis of 12-aryl-8,9,10,12-tetrahydrobenzo[a]xanthene-11-ones [109] (Scheme 23a). For the formation of xanthene, three distinct models were proposed: (a) aldehyde, 5,5-dimethyl-1,3-cyclohexanedione, and 2-naphthol; (b) aldehyde and 2-naphthol; and (c) aldehyde and 5,5-dimehtyl-1,3-cyclohexanedione (Scheme 23b). The data from the substrate scope revealed that aromatic aldehydes with ERG required a longer reaction time than those with EWGs.

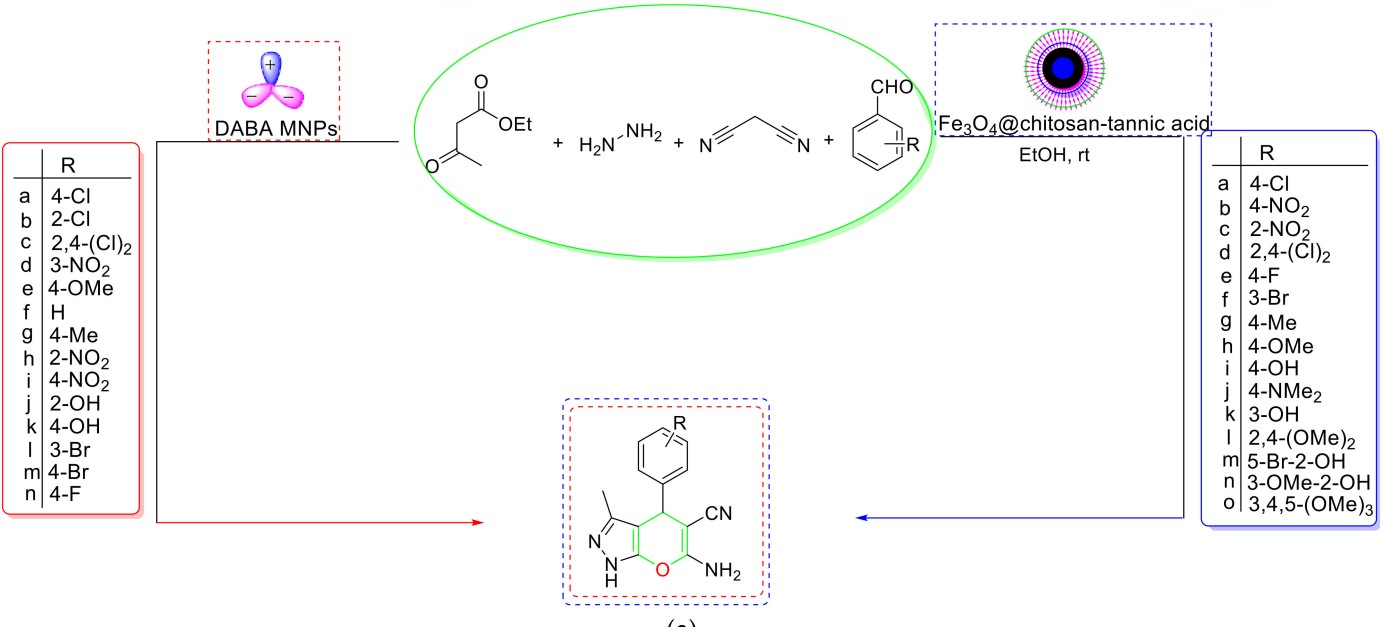

**Scheme 22.** *Cont.*

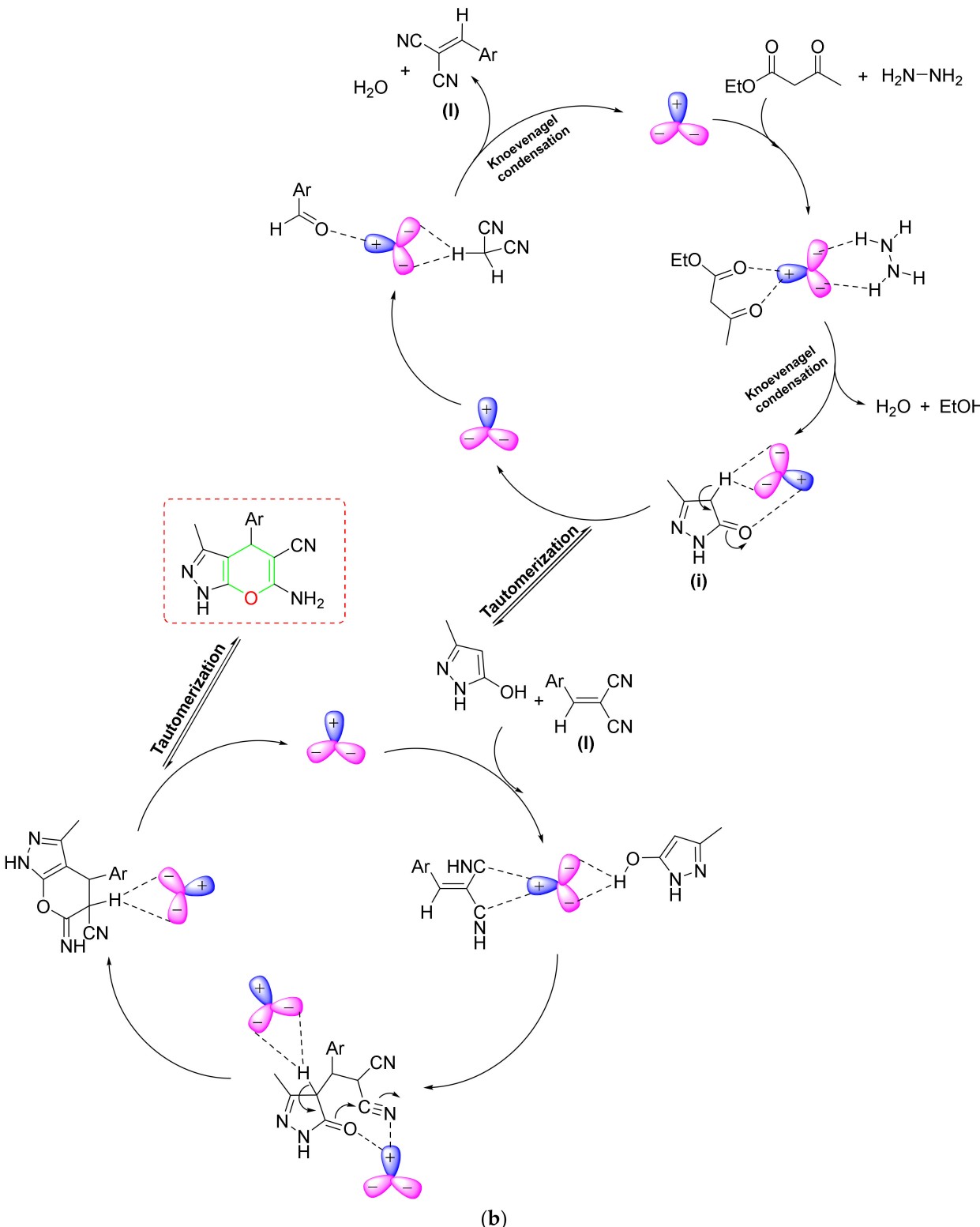

**Scheme 22.** *Cont.*

**Scheme 22.** A schematic presentation of Knoevenagel condensation under different hybrid NCs (**a**); Suggested mechanism for Fe₃O₄@SiO₂@PTS-DABA-nanocatalyzed dihydropyranopyrazole synthesis (**b**); Proposed reaction mechanism for pyranopyrazole synthesis in the presence of Fe₃O₄@chitosan-tannic acid hybrid NC (**c**).

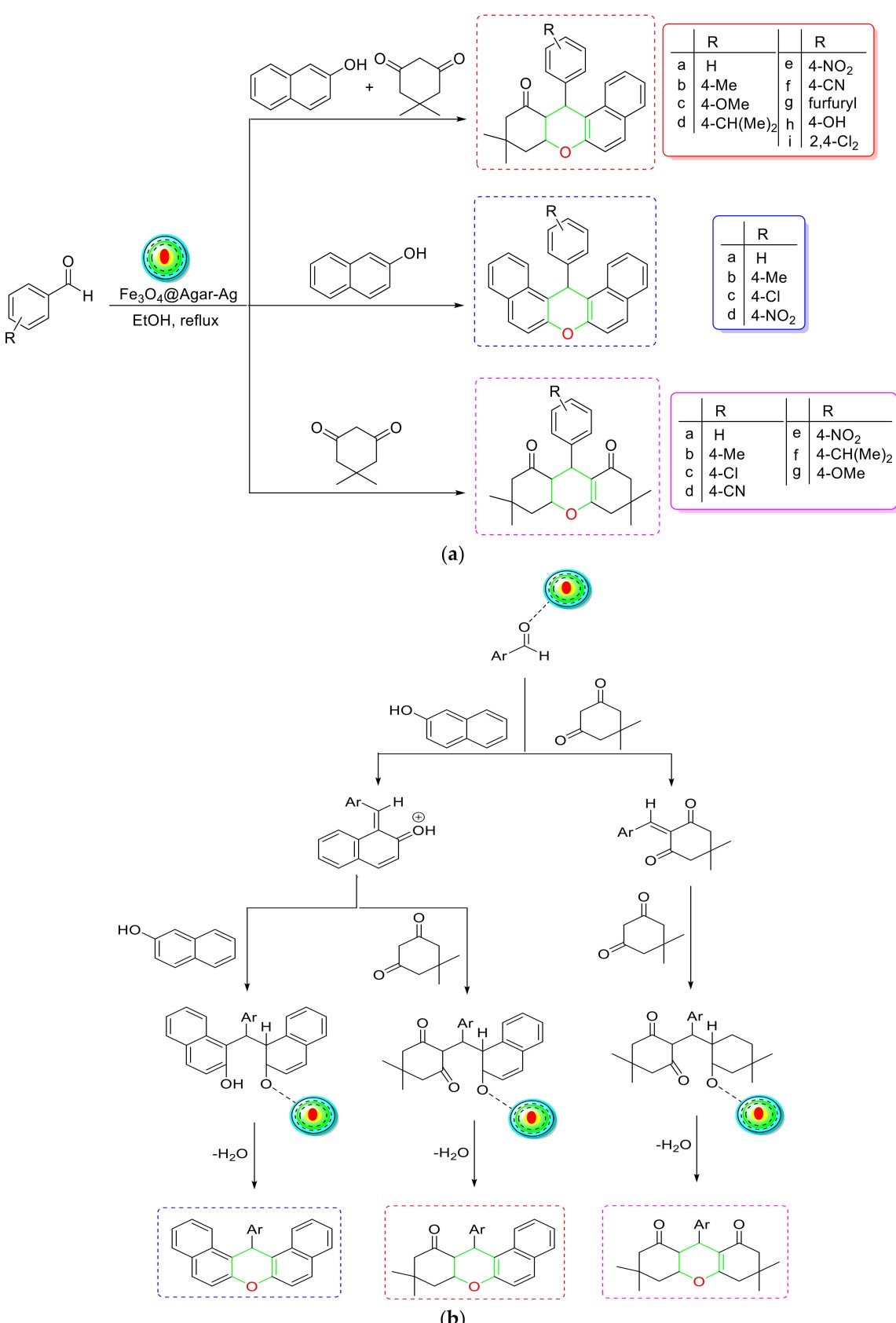

**Scheme 23.** Synthesis of 12-aryl-8,9,10,12-tetrahydrobenzo[a]xanthene-11-ones by utilizing three different models using Fe$_3$O$_4$@Agar-Ag NC (**a**); The possible mechanism for Fe$_3$O$_4$@Agar-Ag-nanoparticle-catalyzed xanthene synthesis (**b**).

Tanuraghaj et al. created a unique sodium-carbonate-functionalized silica-coated-iron oxide NP ($Fe_3O_4@SiO_2@(CH_2)_3OCO_2Na$) as an efficient hybrid NC to manufacture pyrano[2,3-h]-coumarins to avoid NP aggregation or oxidation [110] (Scheme 24a). The mechanistic findings stated that at first, there was deprotonation of OH of 5,7-dihydroxy-4-mehtylcoumarin by the basic catalyst, which produced an intermediate (I) which attacked dialkyl acetylenedicarboxylate and generated a new intermediate (II). After dehydration, a newly created intermediate (II') was attacked with aldehyde, resulting in an adduct (III) that eventually deprotonated, resulting in intermolecular ring closure and the desired product (Scheme 24b). The proposed methodology's key advantages are excellent catalytic activity, good catalyst stability, ten-fold reusability, and mild reaction conditions.

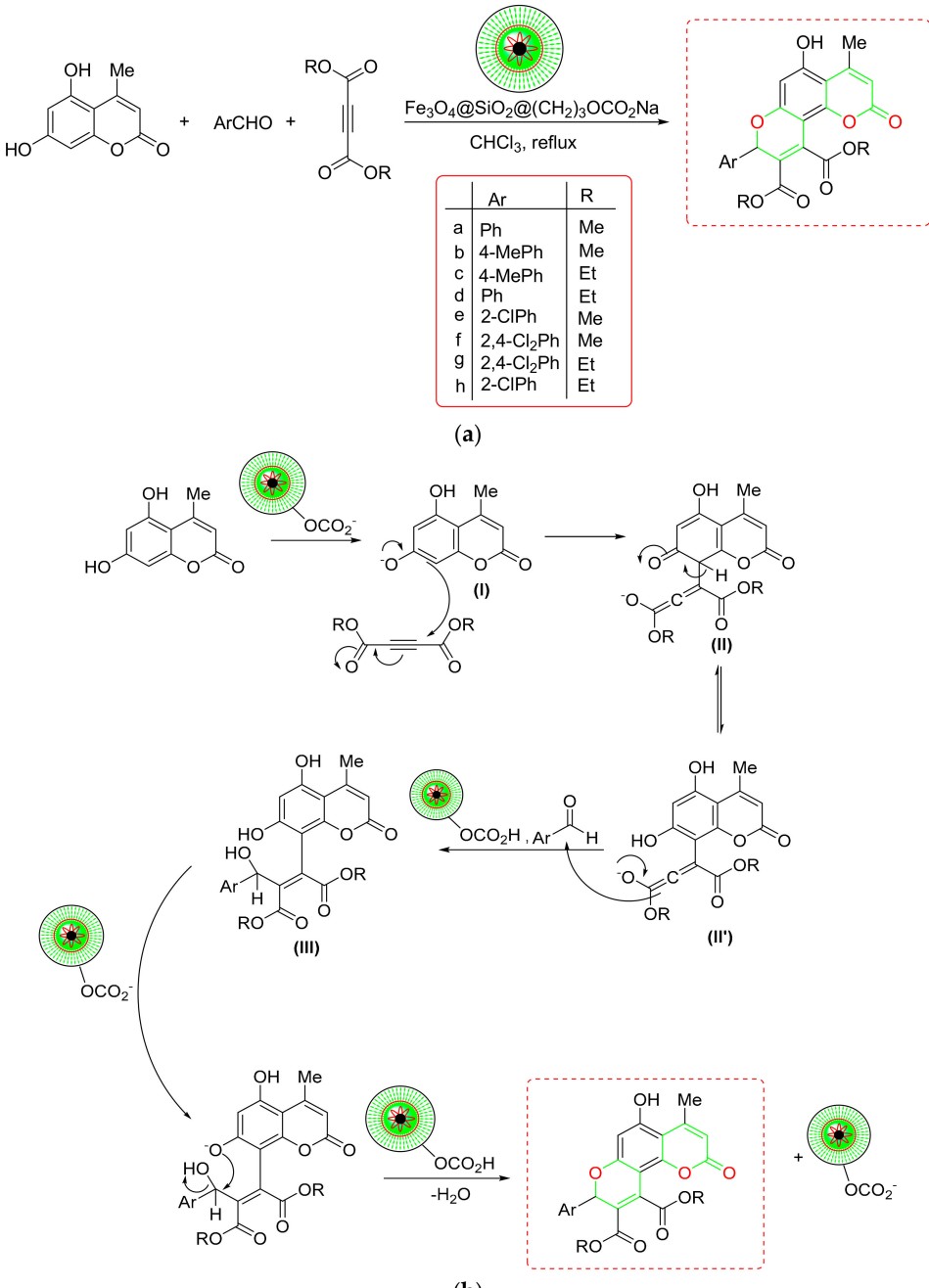

**Scheme 24.** An efficient synthetic route to design pyrano[2,3-h]-coumarins using $Fe_3O_4@SiO_2@$-$(CH_2)_3OCO_2Na$ hybrid NC (**a**); The plausible reaction mechanism for the synthesis of pyrano[2,3-h]-coumarins (**b**).

### 3. Conclusions and Future Perspectives

Green organic transformations are no longer just a notion; rather, they are a critical component of modern organic synthesis. The main goal of green chemistry is to solve the problems that occur with environmental sustainability. The use of metal-based NCs in catalyzing various organic transformations is a highly efficient and greener methodology owing to its enhanced selectivity and higher conversion rates. The last three to four decades have witnessed the importance of these metal-based nanocatalytic systems in reducing the number of steps and purification issues involved in the synthesis of various complex organic molecules. In this review, we have summarized research work by various groups on the synthesis of O-heterocycles (furans, pyrans, coumarins, chalcones, oxazoles, and isoxazoles) using these nanocatalytic species. This review will aid modern researchers in advancing existing synthetic techniques by using multiple metal NCs. The present study aims at recent methods of green nanocatalyzed strategies for designing various O-heterocycles so that researchers working on O-heterocycles can obtain beneficial information on green synthetic tactics for their synthesis. Several reviews on nanocatalyzed organic synthesis are available; however, none of them include the green synthesis of O-heterocycles. As a result, the NC system reported here provides researchers with a reusable technique for efficient O-heterocycle synthesis, operation simplicity, and a green reaction profile.

Although, the field is vastly explored, the race in the designing of various hybrid NCs decorated with various organic species/ligands/polymers/porous skeletons is still in its early stages and holds vast potential, as it will improve the reusability of NCs for a number of repeated cycles in a broad range of solvents, further enhancing selectivity and conversion rates and thereby leading to highly enhanced efficiency of these nanocatalytic systems.

**Author Contributions:** Conceptualization, S.K. (Suresh Kumar), N.K. and M.M.M.-M.; formal analysis and investigation, G.K., E.L. and S.K. (Sanjeev Kumar); writing-original draft preparation, B.S., writing-review and editing B.P., P.R.M., M.K., N.K. and S.K. (Suresh Kumar); Supervision, S.K. (Suresh Kumar) and M.M.M.-M. All authors have read and agreed to the published version of the manuscript.

**Funding:** This research received no external funding.

**Data Availability Statement:** Not applicable.

**Acknowledgments:** The authors gratefully acknowledge Chemistry Department, Kurukshetra University, Kurukshetra; Council of Scientific and Industrial Research (CSIR); and University Grants Commission (UGC), New Delhi for providing support for this work.

**Conflicts of Interest:** The authors declare no conflict of interest.

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
