# Peer review of "Recent Developments in Nanocatalyzed Green Synthetic Protocols of Biologically Potent Diverse O-Heterocycles—A Review"

_catalysts, doi:10.3390/catal12060657_

Round 1
Reviewer 1 Report
The "synthesis" in the title should be changed to "synthetic". The word "diversed" should be also changed to "diverse".
The authors should explain “hybrid metal nanocatalysts” as they have written in the introduction.
The introduction is short and should be extended.
The concept of green chemistry should be explained in the introduction.
After this sentence “a number of review articles on metal nanocatalyzed heterocyclic synthesis have been published to date [37-40]”, more relevant review articles should be cited such as:
https://www.beilstein-journals.org/bjoc/articles/16/52
The authors should compare nano-particles catalyzed reactions with other types of reactions. As the authors can see in the below review article, there are diverse reactions that perform by simple catalysts. There is no need to prepare these nano-catalysts. Please provide at least a sentence about the benefits of nanotechnology by comparing it with simple metal-catalyzed reactions. Please see and cite review articles such as:
https://onlinelibrary.wiley.com/doi/abs/10.1002/aoc.5600
The articles mentioned below should be added to the text.
https://pubs.rsc.org/en/content/articlelanding/2018/ta/c8ta09342c/unauth
https://pubs.rsc.org/en/content/articlelanding/2020/gc/c9gc04040d/unauth
https://dergipark.org.tr/en/pub/jotcsa/issue/62350/904246
https://chemistry-europe.onlinelibrary.wiley.com/doi/abs/10.1002/ejoc.202000999
https://pubs.rsc.org/en/content/articlehtml/2022/nr/d2nr00361a
Section “2.2. Synthesis of Chalcones” is not about O-heterocycles! As the authors mentioned in the title, this review article is about the applications of nanotechnology in the synthesis of O-heterocycles. Please recheck Scheme 6 and Scheme 8.
This article is about the synthesis of benzoxazoles https://pubs.rsc.org/en/content/articlehtml/2022/nr/d2nr00361a. Please add both reaction and mechanism in the section “2.4. Synthesis of Oxazole/Isoxazoles”.
Author Response
We are highly thankful to the learned reviewer for evaluating and recommending the manuscript for revision. A pointwise response to the Reviewer Comments is given below:
Comment 1: The "synthesis" in the title should be changed to "synthetic". The word "diversed" should be also changed to "diverse".
Response 1: Same has been done accordingly.
Comment 2: The authors should explain “hybrid metal nanocatalysts” as they have written in the introduction.
Response 2: We appreciate the suggestion of the learned reviewer and as per the suggestion we’ve introduced hybrid metal nanocatalysts in the introduction part of the article.
Comment 3: The introduction is short and should be extended.
Response 3: As per the learned reviewer’s comment, the introduction section has been modified.
Comment 4: The concept of green chemistry should be explained in the introduction.
Response 4: Thank you for the valuable suggestion. The concept of green chemistry now has been explained at the appropriate place in the manuscript.
Comment 5: After this sentence “a number of review articles on metal nanocatalyzed heterocyclic synthesis have been published to date [37-40]”, more relevant review articles should be cited such as:
https://www.beilstein-journals.org/bjoc/articles/16/52
Response 5: As per the suggestion some more relevant review articles have been cited.
The article mentioned by the learned reviewer has also been cited.
Comment 6: The authors should compare nano-particles catalyzed reactions with other types of reactions. As the authors can see in the below review article, there are diverse reactions that perform by simple catalysts. There is no need to prepare these nano-catalysts. Please provide at least a sentence about the benefits of nanotechnology by comparing it with simple metal-catalyzed reactions. Please see and cite review articles such as:
https://onlinelibrary.wiley.com/doi/abs/10.1002/aoc.5600
The articles mentioned below should be added to the text.
https://pubs.rsc.org/en/content/articlelanding/2018/ta/c8ta09342c/unauth
https://pubs.rsc.org/en/content/articlelanding/2020/gc/c9gc04040d/unauth
https://dergipark.org.tr/en/pub/jotcsa/issue/62350/904246
https://chemistry-europe.onlinelibrary.wiley.com/doi/abs/10.1002/ejoc.202000999
https://pubs.rsc.org/en/content/articlehtml/2022/nr/d2nr00361a
Response 6: As per the suggestion of the learned reviewer few lines comparing nano-catalysed reactions with other types of reactions have been added at the appropriate place and the article suggested were reviewed and have been cited with reference no 29, 30, 45, 46, 91, and 92.
Comment 7: Section “2.2. Synthesis of Chalcones” is not about O-heterocycles! As the authors mentioned in the title, this review article is about the applications of nanotechnology in the synthesis of O-heterocycles. Please recheck Scheme 6 and Scheme 8.
Response 7: We agreed with the reviewer’s comment and in this regard we’ve already mentioned in the respective section (section 2.2) that “Though chalcone does not possess a heterocyclic structure but structurally related to flavonoids and being used as common intermediate for synthesizing various heterocyclic flavonoid type structures and therefore studied along with flavonoids. Therefore, chalcone can be considered as the core part of different O-heterocyclics”. But keeping in mind the reviewer’s comment we’ve revised the section heading from “Synthesis of Chalcones” to “Synthesis of Chalcones- the heterocyclic intermediate”.
Comment 8: This article is about the synthesis of benzoxazoles https://pubs.rsc.org/en/content/articlehtml/2022/nr/d2nr00361a. Please add both reaction and mechanism in the section “2.4. Synthesis of Oxazole/Isoxazoles”.
Response 8: While reading the respective article we’ve found that the article https://pubs.rsc.org/en/content/articlehtml/2022/nr/d2nr00361a is not about the synthesis of benzoxazole rather the article is about “Synthesis of silver and gold nanoparticles enzyme polymer conjugate hybrids as dual activity catalysts for chemoenzymatic cascade reactions”. So we have only cited this reference at the appropriate place.
Please find the point-by-point response in the attachment.

Reviewer 2 Report
The article presents an excellent review of previous work studying the role of metal nanocatalysts in the synthesis of O-heterocycles. A sufficient sampling of the literature is presented in enough detail to aid any "new" researchers in understanding the field (both technically and historically). In this respect, the article can be useful and is suitable for publication.
However, the widespread availability of journal articles does render some of these "literature survey" reviews obsolete. In this regard, the authors should add sufficient commentary (either as a separate section or infused within each section) that provides their perspective of the field. For example, what are the key remaining research questions? Are there new methods required in this field? Any new chemical insights that can be gleaned from this review of the literature?
Indeed, the addition of perspective to "literature surveys" enhance the usefulness of modern review articles as a "map" for future research rather than simply a shortcut through the past.
Author Response
The authors are highly thankful to the reviewer for the critical analysis and recommendation for the article. We admire the comments raised by the learned reviewer.
Response: Thanks for the encouragement and valuable suggestion made by the esteemed reviewer. As per the suggestion now we have addressed the above research questions at different places in the manuscript infused within section and conclusion has been modified to add future perspectives with heading “Conclusion and Future Perspective”.
Please find the point-by-point response in the attachment.
